

# Marine biogeochemical cycling and oceanic CO₂ uptake simulated by the NUIST Earth System Model version 3

Yifei Dai[1, 3], Long Cao[2], Bin Wang[1, 3]

[1] Earth System Modeling Center, and Key Laboratory of Meteorological Disaster of Ministry of Education, Nanjing University of Information Science and Technology, Nanjing 210044, China

[2] Department of Atmospheric Sciences, School of Earth Sciences, Zhejiang University, Hangzhou 310027, China

[3] Department of Atmospheric Sciences and Atmosphere-Ocean Research Center, University of Hawaii, Honolulu HI 96822, USA

*Correspondence: Long Cao (longcao@zju.edu.cn)

**Abstract.** In this study, we evaluate the performance of Nanjing University of Information Science and Technology Earth System Model, version 3 (hereafter NESM v3) in simulating the marine biogeochemical cycle and CO₂ uptake. Compared with observations, NESM v3 reproduces reasonably well the large-scale patterns of upper ocean biogeochemical fields including nutrients, alkalinity, dissolved inorganic, chlorophyll, and net primary production. The model also reasonably reproduces current-day oceanic CO₂ uptake, the total CO2 uptake is 149 PgC from 1850 to 2016. In the 1ptCO₂ experiment, the NESM v3 produced carbon-climate ($\gamma$=-7.9 PgC/K) and carbon-concentration sensitivity parameters ($\beta$=0.8 PgC/ppm) are comparable with CMIP5 model results. The nonlinearity of carbon uptake in the NESM v3 accounts for 10.3% of the total carbon uptake, which is within the range of CMIP5 model results (3.6%~10.6%). Some regional discrepancies between model simulations and observations are identified and the possible causes are investigated. In the upper ocean, the simulated biases in biogeochemical fields are mainly associated with the shortcoming in simulated ocean circulation. Weak upwelling in the Indian Ocean suppresses the nutrient entrainment to the upper ocean, therefore reducing the biological activities and resulting in underestimation of net primary production and chlorophyll concentration. In the Pacific and the Southern Ocean, high-nutrient and low-chlorophyll result from the strong iron limitation. Alkalinity shows high biases in high-latitude oceans due to the strong convective mixing. The major discrepancy in biogeochemical fields is seen in the deep Northern Pacific. The simulated high concentration of nutrients, alkalinity and dissolved inorganic carbon water is too deep due to the excessive deep ocean remineralization. Despite these model-observation discrepancies, it is





expected that the NESM v3 can be employed as a useful modeling tool to investigate large scale interactions between the ocean carbon cycle and climate change.

## 1 Introduction

The global carbon cycle plays an important role in the climate system. The increase in atmospheric carbon

dioxide ($CO_2$) is responsible for a large part of the observed increase in global mean surface temperature (Ciais et al., 2013). From 1750 to 2016, about 645±80 PgC (1 PgC =$10^{15}$ gram carbon) of anthropogenic carbon has been emitted to the atmosphere, including 420±20 PgC from fossil fuels and industry and 225±75 PgC from land-use-change (Le Quéré et al., 2018). This $CO_2$ emission caused atmospheric $CO_2$ concentration to increase by 45% from an annual mean pre-industrial value of ~277 parts per million

(ppm) (Joos and Spahni, 2008) to 406.8 ppm in 2017 (NOAA ESRL Global Monitoring Division, 2017). As a large carbon reservoir, the global ocean contains more than 50 times the amount of carbon than the atmosphere (Denman et al., 2007) and plays a key role in anthropogenic $CO_2$ uptake (Ballantyne et al., 2012; Wanninkhof et al., 2013). Since the year 1870 to 2016, about 25% of anthropogenic $CO_2$ (about 150±20 PgC) has been absorbed by the ocean (Le Quéré et al., 2018).

An increase in atmospheric $CO_2$ perturbs the atmospheric radiative balance and leads to climate change. Changes in atmospheric temperature, precipitation, evaporation, and wind, induce changes in ocean physical properties such as temperature, salinity, and ocean circulation (Gregory et al., 2005; Pierce et al., 2012). These changes in ocean physical properties, in turn, affect the ocean carbon cycle (Sarmiento and Gruber, 2006). For example, increasing sea surface temperature decreases the $CO_2$ solubility and results

in a reduction of oceanic $CO_2$ uptake (Najjar, 1992; Teng et al., 1996; Cox et al., 2000; Zickfeld et al., 2008). Meanwhile, global warming would lead to a weakening of the global thermohaline circulation and an increase in ocean stratification (Gregory et al., 2005; Goris et al. 2015), which would reduce the exchange of carbon and nutrients between the upper ocean and the ocean interior. Global warming would also increase the amount of light in the mixed-layer, and then affect phytoplankton growth and

biologically-mediated $CO_2$ uptake (Polovina et al., 2008; Luo et al., 2009; Steinacher et al., 2010; Capotondi et al., 2012).

Friedlingstein et al., (2006) proposed that the response of oceanic uptake of atmospheric $CO_2$ can be



represented by the linear sum of two components: 1) carbon-concentration sensitivity, which refers to the response of oceanic $CO_2$ uptake to increasing atmospheric $CO_2$; 2) carbon-climate sensitivity, which refers to the response of oceanic $CO_2$ uptake to global warming. Adopting this conceptual framework, a number of studies have analyzed the effect of $CO_2$ concentration and global warming on the carbon cycle

in terms of the carbon-concentration and carbon-climate sensitivity parameters under different $CO_2$ emission and concentration scenarios (Gregory et al., 2009; Boer and Arora, 2009; Tjiputra et al., 2010; Roy et al., 2011; Arora et al., 2013).

Given the importance of carbon cycle feedback in current and future global climate change, it is necessary to include the representation of the global carbon cycle in climate system models. Recently, the third

version of the NUIST earth system model was developed as a registered model of CMIP6 (Cao et al., 2018). NESM v3 consists of three main model components, including European Centre Hamburg Atmospheric Model (ECHAM v6.3) (Stevens et al., 2012; Giorgetta et al., 2013), Nucleus for European Modeling of the Ocean version 3.4 (NEMO v3.4-revision 3814) (Madec and NEMO team, 2012) and Los Alamos sea-ice model version 4.1 (CICE v4.1) (Hunke et al., 2010). NESM v3 has good skills in

simulating internal modes, such as El Niño–Southern Oscillation (ENSO), Madden–Julian oscillation (MJO), and monsoon (Li et al., 2018; Yang and Wang, 2018a; Yang et al., 2018b).

The *Pelagic Interactions Scheme for Carbon and Ecosystem Studies* (PISCES v2) is coupled to the ocean circulation component to represent the ocean biogeochemical processes (Aumont et al., 2015). Séférian et al. (2013) assessed the ability of PISCES from three earth system modes: IPSL-CM4-LOOP, IPSL-

CM5A-LR, and CNRM-CM5.1. The results show that differences in terms of atmospheric component, ocean subgrid-scale physics and resolution would largely influence the marine biogeochemical cycle. Due to the different physical components and multiple non-trivial modifications of physical processes, it is essential to evaluate the performance of the ocean biogeochemical cycle in the NESM v3, i.e. the agreement between simulated and observed fields (Randall et al., 2007).

The outline of the paper is the following. In Section 2, we describe the NESM v3 with a focus on the ocean carbon cycle component, as well as the setup of model simulations. We evaluate the modeled biogeochemical fields against available observations in Section 3.1. In Section 3.2, we evaluate modeled oceanic uptake of anthropogenic $CO_2$ during the historical period against data-based estimates. In Section





3.3, we analyze modeled carbon-concentration sensitivity parameters and carbon-climate sensitivity parameters under different $CO_2$ concentration scenarios and compare our results with CMIP5 model results. Conclusions and discussions are presented in Section 4. We also provide a supplementary material includes comparisons of some biogeochemical fields in the NESM v3 and IPSL-CM5A-LR.

## 2 Method

### 2.1 Model

### 2.1.1 Framework of NUIST-CSM-2.0.1

Detailed model descriptions, major improvements, and tuning are documented in Cao et al. (2018). Here we give a brief introduction.

In this study, we use the low-resolution version of NESM v3. The atmospheric resolution is T31L31 which has a horizontal resolution of ~ 3.75° latitude by 3.75° longitude and 31 layers. The atmospheric model and land surface model are originally adopted from ECHAM v6.3. The detailed information is shown in Stevens et al. (2012) and Giorgetta et al. (2013). The sea-ice component includes four ice layers and one snow layer with a multi-layer thermodynamic scheme (Hunke et al., 2010; Cao et al., 2018). Ocean model runs with the ORCA2 global ocean configuration, which is a type of tripole grid. It is based on a 2 degree Mercator mesh and has 31 layers with the thickness of the ocean layer increasing from 10m in the upper ocean to 500m at 5000m depth. A local transformation is applied in the tropics to refine the resolution to up to 0.5 degree at the equator. In the ocean model, the incoming solar radiation can penetrate to the upper ocean layers as deep as 391m, and a bio-model penetration parameterization scheme is used to calculate the distribution of solar radiation (Lengaigne et al., 2009). In the NESM v3, we modify the ocean background vertical diffusivity to replace the constant value with latitude-dependent values (Jochum et al., 2009, Cao et al., 2018). The parameterization scheme of the vertical diffusivity is detailed in the supplementary and the global distribution of vertical diffusivity is also shown (Fig. S0). Compared to the original vertical diffusivity coefficient constant of 0.12 cm²/s, the coefficients of the tropical ocean are reduced and that of the middle and high latitude oceans are increased, especially in the middle latitude oceans (24°N~33°N and 24°S~33°S). Also, we incorporate the parameterization of brine rejection in the



ocean model based on Smith et al (2010). In the NESM v3, the reference sea-ice salinity is 4 PSU, which means additional salt flux would inject into the ocean during the processes of ice melting.

2.1.2 Ocean biogeochemical component

NESM v3 employs the standard PISCES v2 to represent the ocean biogeochemical cycle. The PISCES

model is developed from a simple Nutrient-Phytoplankton-Zooplankton-Detritus (NPZD) model (Aumont et al., 2002). It can be used for both regional and global simulations of lower trophic levels of the marine ecosystem and ocean carbon cycle (Bopp et al., 2005; Resplandy et al., 2012; Séférian et al., 2013).

In the current version, there are 24 prognostic tracers in total, including dissolved inorganic and organic

carbon, alkalinity, chlorophyll, and nutrients. We use the same biogeochemical parameters as that used in Aumont et al. (2015). The only exception is the advection scheme for passive tracers. Here we use the Total Variance Dissipation (TVD) formulation instead of the Monotone Upstream Scheme for Conservative Laws (MUSCL) formulation to keep the advection scheme to be consistent with the one used in the physical ocean model. Both TVD and MUSCL schemes have good performance in

biogeochemical modeling. The MUSCL scheme has better performance in resolving the small scales, while TVD scheme minimizes systematic error through numerical diffusion, and is a better option for coarse-resolution models (Lévy et al., 2001a).

Two different types of phytoplankton: nanophytoplankton and diatoms, and two size classes of zooplankton: mesozooplankton and microzooplankton, are presented in the model. The life cycle of

phytoplankton is regulated by several processes, including growth, mortality, aggregation, and grazing by zooplankton (Aumont et al., 2015). The growth rate of phytoplankton is determined by temperature, photosynthetic active radiation, and availability of nutrients, including phosphate, nitrate, silicate, iron, and ammonium. The mortality rate of phytoplankton is set as a constant and is identical for nanophytoplankton and diatoms. The aggregations of nanophytoplankton, which transform the dissolved

organic carbon (DOC) to the particular organic matter (POM), only depend on the shear rate, which is set to 1 $s^{-1}$ in the mixed layer and 0.01 $s^{-1}$ below. The same is assumed for diatoms, while the aggregations of diatoms are further enhanced by the nutrients co-limitation. For all species, the phosphate, nitrate, and





carbon are linked by a constant Redfield ratio. In NESM v3, the Redfield ratio of C: N: P is set to be 122:16:1 (Takahashi et al., 1985) and the –O/C ratio is set to 1.34 (Kortzinger et al., 2001). In contrast, the Fe / C, chlorophyll / C, and silicon / C ratio are prognostically simulated by the model predicted based on the external concentrations of the limiting nutrients as in the quota-approach (McCarthy, 1980; Droop, 1983; Aumont et al., 2015).

The remineralization of semi-labile dissolved organic carbon (DOC) can occur in either oxic water or anoxic water depending on the local oxygen concentration, and their degradation rates are specified and identical for oxic respiration and denitrification. Detritus is represented by different types, including particulate organic matter (POM), calcite, iron particles, and diatoms silicate. The POM is described by a simple two-compartment scheme, which uses two tracers corresponding to two size classes: a smaller class (POC: 1-100μm) and a larger class (GOC: 100-500μm). The sinking speed of GOC (50-200 m d$^{-1}$) increases with depth and is much faster than POC (3 m d$^{-1}$). A fraction of phytoplankton would be turned to the POM through the processes of mortality and aggregation. The fate of mortality and aggregation of nanophytoplankton depends on the proportion of the calcifying organisms. For nanophytoplankton, it is assumed that half of the calcifying organisms are associated with the calcifying organisms. Because the density of the calcite is larger than that of organic matter, 50% of the dying calcifiers are routed to the fast-sinking particles. The same is assumed for the mortality of diatoms, and 50% of the dying diatoms are turned to the POM due to the larger density of biogenic silica compared to that of organic matter. The degradation rate of the POM depends on the local temperature with a $Q_{10}$ of about 1.9.

The geochemical boundary condition accounts for the external nutrient supply from five different sources, including atmospheric dust deposition of iron and silicon, river recharge of nutrients, dissolved carbon, and alkalinity, atmospheric deposition of nitrogen, and sediment mobilization of sedimentary iron. At the bottom of the ocean, different sediment parameterization schemes are applied to biogenic silica, POM, and particulate iron. The amount of permanently buried biogenic silica is assumed to balance the external source, the burial efficiency of POM is determined by the organic carbon sinking rate at the bottom follows the algorithm proposed by Dunne et al. (2007), and all the particulate iron would be buried into the sediment once they reach the ocean bottom. The amount of the unburied calcite and biogenic silica would dissolve back into the ocean water instantaneously.



Carbonate chemistry including air-sea $CO_2$ exchange is formulated based on the Ocean Carbon-Cycle Model Intercomparison Project (OCMIP-2) protocol (more information can be accessed at http://ocmip5.ipsl.jussieu.fr/OCMIP/). The quadratic wind-speed formulation proposed by Wanninkhof (1992) is used to compute the air-sea exchange of carbon and oxygen.

## 2.2 Simulations

First, NESM v3 was spun up for 2000 years with all related parameters set to pre-industrial values (the year 1850), including orbital parameters, land use, aerosol, and greenhouse gas (GHGs) concentration (284 ppm for $CO_2$, 790 ppb for $CH_4$, 275 ppb for $N_2O$, and 0 ppt for both $CFC_{11}$ and $CFC_{12}$). The atmosphere and sea-ice components use the end of a 550-year offline simulation as its initial conditions
and the ocean component uses the end of a 4000-year offline simulation as its initial state. Averaged over the last 100 years of the spin-up simulation, linear drift of globally integrated sea-air $CO_2$ flux is 0.0006 PgC yr$^{-1}$ per year, indicating that a quasi-equilibrium state has been reached for the global ocean carbon cycle. Global mean SST averaged over the last 100 years of spin-up simulation is 13.1 Celsius (°C) with the linear drift of -0.0001°C per year, and ocean mean temperature is 3.5 ℃ with the linear drift of
0.00016 ℃ per year, indicating that the dynamic ocean component has also reached a quasi-equilibrium state.

Following the protocol of CMIP6 historical and the shared socio-economic pathway scenarios experiments design (Eyring et al., 2016; Jones et al., 2016), the model is further integrated with changing conditions, including ozone, aerosol, land use, and solar forcing from 1850 to 2100. From the year 1850
to 2014, GHGs concentration and forcing conditions are consistent with observations, and from the year 2015 to 2100, GHGs concentration and forcing conditions are produced based on SSP5-8.5. In addition, following the protocol of CMIP5 (Taylor et al., 2012), we performed an idealized 1%/yr $CO_2$ run (core 6.1 in CMIP5 experiment design, hereafter 1ptCO2), in which atmospheric $CO_2$ concentration increases at a rate of 1% per year starting from the end state of the pre-industrial simulation with other GHGs
concentration remaining at pre-industrial level. The simulation lasted for 140 years until atmospheric $CO_2$ concentration has quadrupled. Also, we conducted a 251-year PI-control simulation.

To separate the effect of atmospheric $CO_2$ and global warming on the ocean carbon cycle, we performed





three types of experiments (biogeochemically coupled, radiatively coupled, and fully coupled). These types of simulations were also performed by previous studies that investigated the effect of $CO_2$ and global warming on the global carbon cycle (Friedlingstein et al., 2006; Arora et al., 2013; Schwinger et al., 2014).

1) Biogeochemically coupled (BC) simulations in which the code of the ocean carbon cycle sees changing atmospheric $CO_2$, but the code of atmospheric radiation sees a constant pre-industrial concentration of $CO_2$. In this way, the ocean carbon cycle is only affected by changing atmospheric $CO_2$, but no direct effect of $CO_2$-induced warming;

2) Radiatively coupled (RC) simulations in which the code of the ocean carbon cycle sees pre-industrial

atmospheric $CO_2$, but the code of atmospheric radiation sees the changing concentrations of atmospheric $CO_2$. In this way, the ocean carbon cycle is only affected by $CO_2$-induced warming, but no direct effect of changing atmospheric $CO_2$.

3) Fully-coupled (FC) simulations in which both the codes of the ocean carbon cycle and atmospheric radiation see the changing concentrations of atmospheric $CO_2$. In this way, the ocean carbon cycle is

affected by changes in both atmospheric $CO_2$ and $CO_2$-induced warming.

In total, there are eight different simulations in this study, including one fully coupled spin-up simulation for 2000 years, one PI-control run (CTRL) for 251 years, three historical+SSP5-8.5 runs (FC, BC, and RC) from 1800 to 2100, and three idealized 1%/yr $CO_2$ runs (FC-1%, BC-1%, and RC-1%) for 140 years.

2.3 Validation data

In this study, we compare the NESM v3 simulated ocean biogeochemical fields, including nutrients, chlorophyll, marine net primary production (NPP), alkalinity, dissolved inorganic carbon (DIC), and oceanic anthropogenic $CO_2$ inventory with available observations and data-based estimates.

Data of global ocean distributions of nutrients concentrations, including nitrate, phosphate, and silicate, are from the *World Ocean Atlas 2018* (WOA18, Garcia, et al., 2018). Geographic distributions of DIC,

alkalinity, and anthropogenic carbon is taken from the *Global Ocean Data Analysis Project v2* (GLODAP) (Key et al., 2015; Lauvset et al., 2016). Both WOA18 and GLODAP v2 data have a horizontal resolution of 1°×1° with 33 levels and represent the climatology in recent decades. We compare modeled chlorophyll





in recent decades with the SeaWiFS dataset (NASA Goddard Space Flight Center, 2014), GlobColour merged data (Maritorena, et al., 2010), and Ocean Colour Climate Change Initiative (OCCCI) merged data (http://www.oceancolour.org/).

Moderate Resolution Imaging Spectroradiometer (MODIS) estimated marine net primary production (NPP) based on three different algorithms are compared with model simulation in this study, including the Standard Vertically Generalized Production Model (VGPM), Eppley-VGPM, and the carbon-based Production Model (CbPM). The datasets can be accessed at http://www.science.oregonstate.edu/ocean.productivity/index.php. In the VGPM and Epply-VGPM, NPP is described as the product of chlorophyll and photosynthetic efficiencies (Behrenfeld and Falkowski, 1997a, 1997b), while the Eppley-VGPM emphasizes the photoacclimation effect at high SSTs (Morel, 1991). In the CbPM, NPP is described as the product of carbon biomass and growth rate (Behrenfeld et al. 2005; Westberry et al. 2008). All three datasets have a horizontal resolution of 1/12°×1/12° from 2003 to 2014. The distribution of observed surface ocean sea-air $CO_2$ flux for the reference year of 2000 is taken from Takahashi et al (2009) and has a spatial resolution of 4° latitude by 5° longitude.

To have a direct comparison between NESM v3 results and observations, we interpolated all modeled results and observations to a 1°×1° grid using the distance-weighted average remapping method, except for the sea-air $CO_2$ flux. Due to the low resolution of observational sea-air flux, we interpolated the modeled result to a 4° × 5° grid.

## 3 Results

### 3. 1 Nutrients

In this section, we compare model-simulated ocean biogeochemical fields, including nutrients, chlorophyll, alkalinity, dissolved inorganic carbon (DIC), and net primary production (NPP), against available observations and data-based estimates.

Nutrients play vital roles in the ocean biogeochemical cycle. A lack of nutrients would limit the growth of phytoplankton. Figure 1 compares the model simulated annual mean spatial distributions of average nutrients concentrations (nitrate, phosphate, and silicate) in the top 100m depth from 1985 to 2014 with the WOA18 observations. The model reproduces reasonably well the large-scale pattern of upper ocean




mean nutrients concentrations. The pattern correlation coefficients (PCCs) of nitrate, phosphate, and silicate are 0.93, 0.91, and 0.83, respectively. The standard deviations (SDs) of nitrate, phosphate, and silicate are 1.05, 1.06, and 1.22, respectively (Fig. 11). Phosphate, nitrate, and silicate in the Southern Ocean have the highest nutrient concentrations of ~1.8, ~25 and ~60 mmol/m$^3$, respectively. Strong

vertical mixing and upwelling bring nutrient-rich deep water to the surface (Whitney, 2011). The relative high nutrients concentration, about 50% of the values in the Southern Ocean, are found in the subarctic Pacific Ocean, and the mid-eastern Pacific Ocean, Relatively low concentrations of nutrients, less than 20% of the values in the Southern Ocean, are found in subtropical regions where the vertical mixing between the surface and the deep ocean is weak.

Some noticeable discrepancies between model simulations and observations are found. Phosphate and nitrate are overestimated in the Southern Ocean and the Pacific Ocean but are underestimated in the Indian Ocean, Subarctic Pacific, and Middle-low latitude Atlantic. Silicate is overestimated nearly over the global ocean, except the Indian Ocean and Subarctic Ocean.

Figure 2 shows the recent 30 years zonal mean latitudinal-depth distributions of nutrients from the FC

simulation and WOA18 observations in the Pacific, the Atlantic, and the global ocean. Nutrients distributions are reproduced well in the Atlantic. The deepest penetration of low-nutrients water to a 1000m depth is simulated in the middle latitude regions. The high concentration of nutrients is found in the Atlantic south of 45°S, and phosphate and nitrate are equator-ward transported by the Antarctic Intermediate Water at near 1000m depth. In the Pacific, the spatial patterns of nutrients broadly agree with

observations, but with noticeable positive biases in the deep Northern Pacific. The simulated centers of phosphate and nitrate-rich water are too deep.

To further analyze the possible reasons for discrepancies in nutrients distribution, we decompose phosphate to its preformed and regenerated components (Weiss et al., 1970; Duteil et al., 2010) and compare the results with the WOA18 observations (Fig. 3). The regenerated phosphate is released through

the remineralization processes of organic matter, and the preformed phosphate is the remaining biotically unutilized surface phosphate, which is transported into the ocean interior by ocean circulation. The regenerated and preformed phosphate are computed as:

$$P_{regenerated} = R_{P:-o_2} \times AOU \tag{1}$$





$$P_{preformed} = P - P_{regenerated} \tag{2}$$

Where AOU is the apparent oxygen utilization, which represents the biological consumption of oxygen. It is computed as the difference between oxygen saturation and simulated oxygen concentration. $R_{P:-o_2}$ represents the oxidation ratio of phosphate and oxygen, which is set to 1/163 in the NESM v3. $P$ represents the simulated phosphate concentration.

For the global, Atlantic, and Pacific Ocean, the preformed phosphate diagnosed from the model accounts for 51%, 47%, and 57% of the total phosphate inventory, and the result diagnosed from the WOA18 is 57%, 55%, and 64%, respectively. A relatively small percentage of the preformed phosphate indicates stronger biological activities in the model. Compared to observations, the model simulates a larger depletion of preformed phosphate (bias is about 0.2 mmol /m$^3$) in the North Atlantic, which indicates too active biological processes in the upper ocean. In the Pacific Ocean, the preformed phosphate concentrations are between 1.3 to 1.5 mmol/m$^3$ from both model simulation and observations.

The regenerated phosphate concentrations have larger variations than preformed phosphate concentration. The noticeable positive biases of regenerated phosphate are found in the deep Northern Pacific. The high regenerated phosphate water in the North Pacific is simulated too deep, and the biases resemble the difference found in latitudinal-depth distributions of nutrients.

In the deep ocean, preformed phosphate is only affected by ocean circulation, while regenerated phosphate is affected by both circulation and remineralization. The NESM v3 simulates the preformed phosphate well but overestimates the regenerated phosphate in the deep ocean, suggesting that the overestimated nutrients in the North Pacific deep ocean are mainly caused by biological processes. Another evidence is that overestimation of nutrients has also been found in other PISCES models, such as IPSL-CM5A-LR (Séférian et al., 2013), while the simulated ocean circulations in IPSL and NESM are different.

We next present the model-simulated pattern of nutrient limitation. In the model, the nutrients limitation coefficient (0~1) is computed from the Michaelis-Menten equation as follow:

$$MM=N / (K+N) \tag{3}$$

Where MM is the Michaelis-Menten coefficient, N is the nutrient concentration, and K is the half-saturation constant.

We calculated the annual mean nutrient limitation coefficient of each nutrient (phosphate, nitrate, silicate,





and iron) and then considered the nutrient with the lowest limitation coefficient as the most limiting factor. Temperature and light are assumed to be the most limiting factor when all nutrients are sufficient for phytoplankton growth and all nutrient limitation coefficients are greater than 0.9. As shown in figure 4, the limiting patterns of nanophytoplankton and diatoms are similar in the mid-low latitude oceans. Iron is the most limiting nutrient for both nanophytoplankton and diatoms in the equatorial Pacific Ocean and the Southern Ocean. Nitrate is the most limiting factor in the subtropical Pacific Ocean, and phosphate is the most limiting factor in the Indian Ocean and middle-low latitude of the Atlantic Ocean. At high latitude oceans, nanophytoplankton is mostly limited by the available light and temperature, while diatoms are mostly limited by silicate. The NESM v3 simulated limiting pattern is generally consistent with the results diagnosed from IPSL-CM4A-LOOP (Schneider et al., 2008), except that the iron limitation diagnosed from the NESM v3 is stronger in the Pacific and Southern Ocean.

## 3. 2 Biological Production

Figure 5 shows the modeled spatial distribution of annual mean surface chlorophyll concentration from 1998 to 2014 compared with Sea-Viewing Wide Field-of-View Sensor (SeaWiFS) observational data (Behrenfeld and Falkowski, 1997a, 1997b), GlobColour merged data, and Ocean Colour Climate Change Initiative (OCCCI) merged data.

In the NESM v3, chlorophyll in both nanophytoplankton and diatoms are parameterized based on the photo-adaptive model (Geider et al., 1997) in which chlorophyll is regulated by the chlorophyll-to-carbon ratio, growth of plankton biomass, mortality, aggregation, and the grazing by zooplankton. The large-scale pattern of simulated ocean chlorophyll concentration broadly agrees with observations with high levels of chlorophyll in the subarctic Pacific Ocean, North Atlantic, equatorial Pacific, and the Southern Ocean and low levels of chlorophyll in the subtropical oceans. The relatively high chlorophyll concentrations along the extratropical coastal regions are reproduced, but the model generally underestimates chlorophyll concentration in the tropical coastal regions, especially in the tropical Indian Ocean, maritime continent, and the tropical Atlantic Ocean. This underestimation is partly associated with the deficiencies in modeled coastal dynamics, which is usually not represented well by the relatively coarse global ocean models (Aumont et al., 2015). It is reported that the observed chlorophyll distribution





is better reproduced when PISCES is coupled to a higher resolution ocean circulation model (Lee et al., 2000; Hood et al., 2003; Kone et al., 2009). Also, we can see an underestimation of chlorophyll over the entire Northern Indian Ocean. This is associated with the underestimation of nutrients over the Indian Ocean (Fig. 1) that increase nutrients limitation and inhibit growth.

In the Southern Ocean where the seawater is typically characterized by high nutrients and low chlorophyll (Lin et al., 2016), noticeable discrepancies are seen among different observational datasets that are associated with different algorithms used for different products. For example, in the intermediate concentration regions such as the Southern Ocean, chlorophyll derived from reflectance by standard algorithms tend to be underestimated by a factor of about 2 to 2.5 (Kahru and Mitchell, 2010). In the

Southern Ocean, the NESM v3 overestimates the chlorophyll concentration in the east of 150°E and underestimates it near the International Date Line. In the Atlantic part of the Southern Ocean, the modeled chlorophyll concentration is within the range of observational estimates, higher than SeaWiFS but lower than GlobColour and OCCCI.

Figure 6 shows the annual mean climatology of vertically integrated NPP from 1998 to 2014. Three

different algorithms, including VGPM, Epply-VGPM, and CbPM, are used to estimate the NPP based on the MODIS observation data. Both VGPM and Epply-VGPM are chlorophyll-based algorithms in which NPP is calculated as a function of chlorophyll, available light, and the photosynthetic efficiency. The only difference between VGPM and Epply-VGPM is the description of photosynthetic efficiency (Behrenfeld and Falkowski, 1997a, 1997b). The Eppley-VGPM emphasizes the effect of SST, i.e. growth rate is higher

at high temperature regions (Eppley, 1972). Therefore, compared to the VGPM, the Eppley-VGPM estimates more NPP in low latitude oceans and less at high latitude oceans (Fig. 6b and 6d). NPP in the CbPM is described as the product of carbon biomass and growth rate (Behrenfeld et al. 2005; Westberry et al. 2008). In the NESM v3, the NPP is also described as the product of phytoplankton biomass and growth rate, although the calculation of growth rate in the NESM v3 is more complex, which involves

chlorophyll, nutrients availability, temperature, respiration, and the photosynthetically active radiation (PAR). Compared to the three products, the climatology of the NESM v3 simulated vertically integrated NPP resembles Epply-VGPM and CbPM estimates. High level of NPP (more than 250 g C/m$^2$/year) in the eastern equatorial Pacific and middle-latitude oceans around 40°S and 40°N and the low level of NPP



(less than 100 g C/m$^2$/year) in the middle-low latitude oceans, the Southern Ocean, and high latitude oceans is reproduced. Also, the high level of NPP (more than 325 g C/m$^2$/year) in low latitude coastal regions is reproduced to some extent.

Although the global pattern of NPP broadly agrees with the observational estimates, PCC between model simulation and Epply-VGPM is only 0.5, indicating that some local features are not well described in the NESM v3. Compared to CbPM and Epply-VGPM, the NESM v3 significantly underestimates the NPP in the Indian Ocean. The NESM v3 also underestimates the NPP in the eastern coastal areas of the United States and the Arctic coastal areas.

Averaged from 2003 to 2014, the globally integrated ocean NPP from the NESM v3 simulation is 45.1 PgC yr$^{-1}$, compared with the data-based estimates of 37 to 67 PgC yr$^{-1}$. The large range of data-based estimates of global NPP is a result of different satellite observations and different algorithms for the NPP estimation (Longhurst et al., 1995; Antoine et al., 1996; Behrenfeld and Falkowski, 1997b; Behrenfeld et al., 2005). Global NPP simulated by CMIP5 models also shows a wide range of values from 30.9 to 78.7 PgC yr$^{-1}$ (Bopp et al., 2013). NESM v3 simulated global NPP is within the range of data-based estimates and current CMIP5 model estimates. Of the NESM v3 simulated global ocean NPP, 20% is contributed by diatoms, and 80% is contributed by nanophytoplankton. For comparison, from the data-based estimate, 7% to 32% of the total NPP is associated with diatoms (Uitz et al.,2010; Hirata et al., 2011), while ocean biogeochemical models estimate that 15% to 30% global NPP is from diatoms (Aumont et al., 2003; Dutkiewicz et al., 2005; Yool et al., 2011).

3. 3 Dissolved inorganic carbon and alkalinity

Figures 7 and 8 display the modeled and observed alkalinity and DIC averaged over the upper ocean (0-100m) and along zonally averaged section in the Pacific Ocean, Atlantic Ocean, and the global ocean. The model's skills in simulating alkalinityare moderate (PCC = 0.56). The observed high alkalinity in the subtropical surface oceans and low alkalinity near the maritime continent are simulated and the modeled global upper ocean mean alkalinity only has a minor negative bias of 0.45%. The major discrepancies are seen in the Southern Ocean and the subarctic Pacific with a positive bias of more than 80 mmol/m$^3$. In high-latitude oceans, convective mixing of alkalinity-rich deep water is an important factor of changing

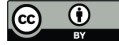



upper ocean alkalinity, and SST can be used as a proxy of the convective mixing change (Lee et al., 2006). An underestimation of SST of 1℃ is simulated at high latitude oceans (figures not shown), indicating a stronger convective mixing, which may explain the overestimated alkalinity at high latitude oceans. The alkalinity has a negative bias of more than 60 mmol/m$^3$ near the maritime continent, where the alkalinity

concentration is usually related to salinity (Lee et al., 2006). Cao et al. (2018) found that the underestimation of surface salinity of 2 PSU is caused by excessive precipitation in this region.

NESM v3 simulates well the large-scale pattern of the observed DIC (PCC = 0.78) with high DIC concentrations in the middle-high latitude Atlantic and low DIC concentrations in the middle-low latitude Pacific and the entire Indian Ocean. The model simulated global upper oceanmeans DIC only has a minor

positive deviation of 0.27%. Although the global pattern of DIC is different from alkalinity, their deviation patterns are similar. A positive DIC bias of more than 80 mmol C/m$^3$ is seen in the Southern Ocean and negative bias of more than 40 mmol C/m$^3$ is seen in the maritime continent.

The large-scale patterns of the zonal averaged latitudinal-depth distribution of both DIC and alkalinity are simulated well in the Atlantic Ocean. Apparent biases of DIC and alkalinity are seen in the deep

Northern Pacific. One noticeable pattern of observed DIC and alkalinity distributions is that their maximum concentrations are around 2000-3000m of the North Pacific Ocean, which the model fails to reproduce. The model also overestimates DIC storage in the deep Pacific Ocean. The mismatches between model simulation and observations, i.e. underestimation of DIC and alkalinity concentrations in the upper 1000m depth and overestimation of their concentrations in the deep ocean, resemble those of nitrate and

phosphate. It indicates that model-data discrepancies of alkalinity and DIC may also be attributed to excessive deep and active remineralization processes, which would release a large amount of dissolved carbon in the deep ocean.

### 3.4 Oceanic CO$_2$ uptake

In this section, we compare the NESM v3 simulated anthropogenic carbon uptake during the historical

period (FC) against available observations.

First, we compare the NESM v3 simulated sea-air CO$_2$ flux against available observations for the reference year of 2000 (Takahashi et al., 2009). As shown in Fig. 9, the NESM v3 realistically reproduces





the large-scale pattern of observed sea-air $CO_2$ flux with $CO_2$ outgassing in the equatorial oceans and uptake in the mid-to-high latitude oceans (PCC=0.71 and SD=1.04). For both observation and model results, strong $CO_2$ uptake is found in the North Atlantic where sea surface temperature is low and the formation of deep water is active. Compared to the data-based estimates, there are overestimates of modeled sea-air $CO_2$ flux in the tropical Pacific, near 45°S oceans, and near 30°N oceans, and the strongest underestimates of modeled sea-air CO2 flux are seen in the high-latitude oceans (Fig. 9c and 9d). The globally integrated ocean uptake flux from observation is 2.0 ± 0.7 PgC in the year 2000 (Takahashi et al., 2009), while the value is 2.8 Pg C from the model simulation. The deviation is mainly originated from positive bias in the pre-industrial steady-state oceanic $CO_2$ uptake due to the 3-dimensional correction of nutrient and alkalinity in the PISCES model (Séférian et al., 2015; Aumont et al., 2015), In the NESM v3, the pre-industrial steady-state of total oceanic $CO_2$ uptake is 1.0 Pg C per year, compared with the observation value of 0.4 ± 0.2 Pg C per year.

We compared the NESM v3 simulated anthropogenic $CO_2$ budget with the data-based estimate provided by the Intergovernmental Panel on Climate Change Fifth Assessment Report (IPCC AR5) (Table 1). The model-simulated ocean uptake of anthropogenic $CO_2$ is slightly lower than that from the IPCC AR5 but within the estimated uncertainty range. From the pre-industrial time to the year 2011, NESM v3 simulated cumulative oceanic $CO_2$ uptake is 137.2 PgC, compared with data-based estimates of 155 ± 30 PgC. The decade average oceanic anthropogenic CO2 uptake diagnosed from the FC run increases from 1.7 to 2.3 PgC yr$^{-1}$ from 1980 to 2009, while the observation ranges from 2.0±0.7 to 2.4±0.7 PgC yr$^{-1}$. Also, compared to recent results (Le Quéré et al., 2018), from the year 1870 to 2016, the modeled cumulative CO2 uptake of 149 PgC is within the range of 150 ± 20 PgC.

The vertically integrated column inventory of modeled ocean storage of anthropogenic DIC (i.e. FC minus CTRL simulation) from 2000 to 2004 (Fig. 10a) and from 1992 to 1996 (Fig. 10c) are compared with GLODAP v2 (Fig. 10b) and GLODAP v1 (Fig. 10d), receptively. NESM v3 reasonably captures the large-scale data-based distribution of anthropogenic DIC. The largest inventory in the 2000s of more than 100 mol C m$^{-2}$ is simulated in the Northern Atlantic where SST is low and deep water formation is active. In the model simulation, the North Atlantic stores 20.8% of the global oceanic anthropogenic carbon, while it is 17.6% in the observation (Fig. 11a and 11b). In other oceans, the large inventory is mainly found in





the middle-latitude areas near 30°N and 30°S. In the Southern Hemisphere Oceans, 58.9% of the global oceanic anthropogenic DIC inventory is simulated, compared to the value of 62.6% in the observation. The most noticeable discrepancy between the GLODAP v2 and model simulation around 2002 is found in the south of 50°S. Only 8.3% of the global oceanic anthropogenic DIC inventory is stimulated, while

the value is 15.5% in the observation. However, we noticed that the vertically integrated anthropogenic DIC concentration is also low in the southern ocean south of 50°S in the GLODAP v1 and only 9.9% of the global inventory is stored in this region. The oceanic anthropogenic DIC storage is the cumulative result of the air-sea $CO_2$ exchange. Takahashi et al. (2009) found the $CO_2$ outgassing amount of about 1 mol C m$^{-2}$ yr$^{-1}$ in the Southern Ocean south of 45°S. In this aspect, the storage in this region should be as

small as that in the GLODAP v1 and the simulated results. It is noted that anthropogenic DIC in the GLODAP is diagnosed by a relative crude application of the transit time distribution method, and thus the results are subject to considerable uncertainties (Lauvset et al., 2016).

Figure 11 shows the zonal mean latitudinal-depth distribution of anthropogenic DIC concentration in the Atlantic, Pacific, and the global Ocean from the NESM v3 FC simulation and GLODAP v2. The

anthropogenic $CO_2$ invades the ocean in the air-sea interface and then penetrates downward. The observed highest concentrations (more than 51 mmol C m$^{-3}$) in near-surface waters and the observed low concentration (less than 3 mmol C m$^{-3}$) in most of the deep ocean (the Pacific and the middle-low latitude Atlantic) are simulated. For both data-based estimates and model simulations, a substantial amount of anthropogenic $CO_2$ has penetrated down to the ocean interior as deep as 1000 m depth with two

penetration tongues near 30°N and 40°S and the deepest penetration of anthropogenic DIC is found in the Northern Atlantic. Deep penetration of anthropogenic DIC is typically associated with convergence zones at temperate latitudes and high latitude oceans where vertical mixing is strong (Sabine et al., 2004). Similar to the vertically integrated inventory, the major discrepancy in the latitudinal-depth distribution is also found in the Southern Atlantic south of 50°S.

Figure 12 compares the spatial pattern of the NESM v3 simulated biogeochemistry-related fields with corresponding observations using a Taylor diagram (Taylor, 2001). In summary, model-simulated statistical patterns of the upper ocean nutrients compare well with observations, while the simulated spatial patterns of chlorophyll, primary production, and alkalinity show larger deviations from





observations. It is noted that chlorophyll and NPP are not directly observed but diagnosed from the observation-based data, and thus their estimations are subject to considerable uncertainties.

3. 5 Response of the oceanic $CO_2$ uptake to atmospheric $CO_2$ and global warming

Increasing atmospheric $CO_2$ affects oceanic $CO_2$ uptake directly. Meanwhile, global warming also affects the ocean carbon cycle via changes in climatic fields such as temperature and ocean circulation. In this section, we first presented the NESM v3 simulated physical climate change and oceanic $CO_2$ uptake under the historical and SSP5-8.5 scenario. Then, we presented NESM v3 simulated oceanic $CO_2$ uptake and carbon cycle sensitivity parameters in the 1ptCO₂ runs and compared the simulated results with that from CMIP5 models.

3. 5. 1 NESM v3 simulated physical climate change under historical and SSP5-8.5 scenario

Figure 13 shows the NESM v3 simulated changes (minus control simulation) in global annual mean surface air temperature (SAT), mixed layer depth (MLD), and the intensity of Atlantic meridional overturning circulation (AMOC) at 30°N from 1850 to 2100 under the historical and SSP5-8.5 scenario. In the FC simulation, the annual global mean SAT anomaly averaged over the period of 2080 to 2100 (relative to the period of 1986-2005) is 4.6 K, which is at the higher end of the CMIP5 model results (2.6 ~ 4.7 K) under the RCP 8.5 scenario (Collins and Knutti, 2013; Knutti and Sedláček, 2013). It is noted that the CMIP6 input forcing is used in this study and the atmospheric $CO_2$ concentration at the end of the 21$^{st}$ century in SSP5-8.5 is about 10% higher than the concentration in the CMIP5 RCP 8.5 scenario. With the increasing atmospheric temperature, the global ocean also becomes warmer in FC and RC simulations, reducing $CO_2$ solubility and acting to mitigate oceanic $CO_2$ uptake.

MLD is seen decreasing since the 1980s. The reduction of mixed layer depth, which is associated with a relatively faster warming of the surface ocean and a slower response of the deep ocean, indicates a more stratified upper ocean with global warming (Held et al., 2010). A substantial weakening of AMOC intensity in the RC and FC simulations is seen in the 21$^{st}$ century, which is associated with ocean surface warming and increased freshwater input into the North Atlantic (Gregory et al., 2005). In the pre-industrial period, the model-simulated AMOC index at 30°N is 17.5 Sv (1Sv $=10^6$ m$^3$ s$^{-1}$), within the range from 14





to 31 Sv from CMIP5 models (Weaver et al., 2012). The modeled annual mean of AMOC transport at 30°N averaged from 2004 to 2011 is 17.1 Sv, while the observation record during the same period from RAPID/MOCHA (Rapid Climate Change programme / Meridional Ocean Circulation and Heatflux Array) is 17.5 ± 3.8 Sv (Rhein et al., 2013). By 2100, the simulated intensity of AMOC declines to 8.0 Sv. The

simulated 54% weakening of AMOC by the end of this century is at the higher end of what is simulated by CMIP5 models that range from 15% to 60% under the RCP 8.5 scenario (Cheng et al., 2013). The higher atmospheric $CO_2$ concentration at the end of 2100 in the SSP5-8.5 may partly explain the larger AMOC change in this study. Also, Cao et al. (2018) pointed out that the equilibrium climate sensitivity to $CO_2$ forcing in the NESM v3 is about 10% higher than the CMIP5 ensemble.

3. 5. 2 NESM v3 simulated oceanic $CO_2$ uptake under historical and SSP5-8.5 scenario

The ocean carbon cycle is regulated by changes in atmospheric $CO_2$ and physical climate (Doney et al., 2004). In the FC simulation, weakening of the vertical ocean mixing, as indicated by the reduced mixed layer depth, will reduce the vertical transport of $CO_2$ from the upper ocean to ocean interior, and the weakening of AMOC would significantly reduce the oceanic $CO_2$ uptake in the Northern Atlantic (Roy

et al., 2011). A warmer surface ocean would reduce $CO_2$ solubility, also reducing oceanic $CO_2$ uptake. Figure 14 shows the time evolution of the oceanic $CO_2$ uptake from the BC, RC, FC, and the linear sum of BC and RC. In the BC simulation, the global ocean absorbed a total of 662 PgC of anthropogenic $CO_2$ from the atmosphere by the year 2100. In the RC simulation, the increased sea surface temperature, enhanced ocean stratification, and the weakened AMOC all act to decrease $CO_2$ uptake. As a result, global

warming alone causes the ocean to release $CO_2$ into the atmosphere. By the year 2100, the modeled cumulative $CO_2$ uptake is -35.9 PgC. In the FC simulation, oceanic $CO_2$ uptake is affected by both the increase in atmospheric $CO_2$ and global warming. By the end of the 21st century, simulated cumulative oceanic $CO_2$ uptake since the pre-industrial era is 567 PgC, which is within the ranges from 420 PgC to 600 Pg C from CMIP5 models results under the RCP 8.5 scenario (Jones et al., 2013).

The sum of the simulated oceanic $CO_2$ uptake from the BC and RC simulations (626 PgC) is larger than that from the FC run (567 PgC), indicating that the effect of increasing atmospheric $CO_2$ (carbon-concentration sensitivity) and the effect of global warming (carbon-climate sensitivity) on the oceanic



$CO_2$ uptake is not exactly additive. This nonlinearity was also found in previous studies (Boer and Arora, 2009; Gregory et al., 2009; Schwinger et al., 2014). The NESM v3 simulated nonlinearity (i.e., BC+RC-FC) is 59 PgC by the end of the 21st century. This nonlinearity is about 10.4% of the total ocean uptake, and it is larger than the absolute value of the radiative effect on ocean carbon uptake (-35.9 PgC).

To better understand oceanic $CO_2$ uptake in response to changing atmospheric $CO_2$ and global warming, Figure 15 shows the spatial distribution of anthropogenic sea-air $CO_2$ flux at the end of the 21st century (averaged over the year 2091 to 2100) under the SSP5-8.5 scenario from FC, RC, BC simulations, and the difference between FC simulation and the sum of RC and BC simulations.

In the BC simulation, the total oceanic anthropogenic $CO_2$ uptake is 8.0 Pg C at the end of the 21st century.

The ocean absorbs atmospheric $CO_2$ in most regions except for a few scattered grid points at the mid-latitudes with slight $CO_2$ outgassing. The strongest $CO_2$ uptake of about 150 g C m$^{-2}$ yr$^{-1}$ is found in the North Atlantic, subarctic Pacific, and the Southern Ocean between 45°S and 60°S. Results from the RC simulation show $CO_2$ outgassing in large parts of the global ocean as a result of global warming that reduces the $CO_2$ solubility and increases the oceanic $pCO_2$. The total $CO_2$ outgassing in the RC simulation

is 0.67 Pg C, less than 10% of the amount of $CO_2$ uptake in the BC simulation. The warming also has a direct impact on the marine biological processes by altering metabolic, photosynthesis, and respiration rates of plankton. As a consequence, the changes in biological production and the subsequent export of organic matter and $CaCO_3$ changes may further affect the oceanic $CO_2$ uptake by altering the DIC, alkalinity, and biological pump (Olonscheck et al., 2013; Lewandowska et la., 2014). Plattner et al. (2001)

found that the biologically mediated changes enhance ocean $CO_2$ uptake at the high latitude, and reduce ocean $CO_2$ uptake at the low latitude. Cao et al. (2017) found that 20% of warming reduced cumulative oceanic $CO_2$ uptake is associated with the change in marine biological rates. However, this result has a large model dependency, and the net biological effect on $CO_2$ uptake is uncertain because of the complex interaction among biological activities. In the Arctic Ocean, warming induces a net uptake of $CO_2$ of 0.07

PgC yr$^{-1}$ (~6 g C m$^{-2}$ yr$^{-1}$) because the reduced sea-ice extent under global warming allows more open seawater to absorb atmospheric $CO_2$. The FC simulation shows the combined effect of increasing atmospheric $CO_2$ and global warming on the oceanic $CO_2$ uptake (Fig. 14c). Oceanic $CO_2$ uptake is simulated in most regions, indicating the dominant role of the increasing atmospheric $CO_2$ on the oceanic





carbon uptake. Similar to the BC simulation, the strongest $CO_2$ uptake is simulated in the Southern Ocean. Due to the reduced AMOC, the capacity of the Northern Atlantic uptakes $CO_2$ is significantly suppressed. Thus, some regions of the Northern Atlantic even appear $CO_2$ outgassing. Also, $CO_2$ outgassing is seen in the subtropical Pacific, indicating that the radiative effect dominates the biogeochemical effect in this

region.

Figure 15d shows the spatial distribution of differences in sea-air $CO_2$ flux between the FC simulation and the sum of the BC and RC simulations during the 2090s. The differences represent the nonlinearity between carbon-climate sensitivity and carbon-concentration sensitivity. In the NESM v3, a relatively large nonlinearity is simulated in the Northern Atlantic north of 45°N (19.8% of the total nonlinearity)

and the Southern Ocean south of 40°S (35.3% of the total nonlinearity), which is consistent with the findings of previous studies (Zickfeld et al., 2011; Schwinger et al., 2014). The background simulation effects can partly explain the nonlinearity. Compared with the radiatively coupled simulation, more carbon is subject to the impact of climate change in fully coupled simulations. As a consequence, in fully coupled simulations, the increased temperature would have a larger effect on $CO_2$ solubility and buffer

factor (Yi et al., 2001). Also, reduced ocean circulation and increased ocean stratification would slow down the transport of anthropogenic $CO_2$ from the surface to the deep ocean. Thus, compared to the BC simulation, slowing ocean ventilation in FC would cause a larger reduction in oceanic $CO_2$ uptake. The oceanic carbon uptake in the fully-coupled simulations is lower than the sum of the BC and RC simulations, which is consistent with other CMIP5 models (Schwinger et al., 2014).

3. 5. 3 carbon-concentration and carbon-climate sensitivity parameters diagnosed from the NESM v3.

In this section, we investigate oceanic $CO_2$ uptake under the framework of the carbon-concentration and carbon-climate sensitivity parameters.

Arora et al. (2013) diagnosed these two parameters from two types of experiments performed by a subset of CMIP5 models, i.e., BC simulations and RC simulations.

In the biogeochemically-coupled simulations where the ocean carbon uptake is only affected by changing atmospheric $CO_2$. The relationship between atmospheric $CO_2$ concentration and sea-air $CO_2$ flux can be simplified as:



$$\int_0^t F' \, dt \approx \beta \Delta C_A \tag{5}$$

Where $F'$ represents oceanic carbon uptake change in the biogeochemically coupled simulation. In the radiatively-coupled simulations where the oceanic carbon uptake is only affected by temperature change. The relationship between temperature and sea-air $CO_2$ flux can be simplified as:

$$\int_0^t F' \, dt \approx \gamma \Delta T \tag{6}$$

Where $F'$ represents oceanic carbon uptake change in the radiatively coupled simulation.

In this study, we estimate the strengths of sensitivity parameters of carbon-concentration and carbon-climate sensitivities, using equations (5) and (6). Figure 16 shows the change in ocean carbon storage against the change in the atmospheric $CO_2$ concentration (Fig. 16a) and the global annual mean surface temperature (Fig. 16b), respectively. The derived evolution of the carbon-concentration sensitivity parameter $\beta$ as a function of atmospheric $CO_2$ concentration and carbon-climate sensitivity parameter $\gamma$ as a function of the change in temperature is shown in Fig. 16c and 16d, respectively.

As shown in Fig. 16, in the BC and RC simulations, modeled ocean storage of anthropogenic $CO_2$ scales roughly linearly with atmospheric $CO_2$ and changes in global mean surface temperature. Increasing atmospheric $CO_2$ alone increases oceanic $CO_2$ uptake whereas increasing temperature alone decreases $CO_2$ uptake. Therefore, the carbon-climate parameter $\gamma$ is negative while the carbon-concentration parameter $\beta$ is positive. In the year 2100, the carbon-climate parameter is -5.4 Pg C/K and the carbon-concentration parameter is 0.79 Pg C/ppm. From 1850 to 2100, the carbon-climate parameter decreases with the increasing temperature change, indicating that with enhanced warming, each degree of surface temperature increase would induce more $CO_2$ outgassing from the ocean (Fig. 16d). The Carbon-concentration parameter initially increases with atmospheric $CO_2$ and then decreases (Fig. 16c). The decreasing trend of $\beta$ is consistent with the slowdown of the increasing trend of the oceanic $CO_2$ uptake at the end of the 21st century as a result of decreased oceanic buffer ability due to the increasing DIC concentration. Similar trends of carbon-climate and carbon-concentration sensitivity parameters are also found in previous studies (Arora et al., 2013). The increased sensitivity of $CO_2$ outgassing to temperature and the decreased sensitivity of $CO_2$ uptake to atmospheric $CO_2$ concentration indicate that the ocean's ability to absorb atmospheric $CO_2$ would be weakened with





increasing atmospheric $CO_2$ and global warming.

3. 5. 4 Carbon-concentration and carbon-climate sensitivity parameters from 1ptCO₂ runs.

Arora et al., (2013) analyzed carbon-concentration and carbon-climate sensitivity parameters from CMIP5 models using the benchmark simulations in which atmospheric $CO_2$ is assumed to increase at a rate of 1% per year for 140 years to reach 4×$CO_2$. It is reported that the carbon feedback parameters of β and γ are sensitive to $CO_2$ scenarios (Gregory et al., 2009; Arora et al., 2013). To have a direct comparison with CMIP5 results, we performed a similar set of simulations.

The total $CO_2$ uptake during the 140 years in FC-1% is 636 Pg C, while the results from CMIP5 models range from 533 to 676 Pg C. The sum of the total $CO_2$ uptake in the RC-1% and the BC-1% is 65.7 Pg C larger than that in the FC-1%. The simulated nonlinearity (i.e. BC-1% + RC-1% - FC-1%) is about 10.3% of the total $CO_2$ uptake in the FC-1%, which is at the higher end of the nonlinearity estimated by CMIP5 models range from 3.6%-10.6% (Schwinger et al., 2014).

Then, we compare NESM v3 simulated β and γ parameters with those of CMIP5 results. Figure 17 shows the simulated β and γ parameters in the 1ptCO₂ runs. At the end of 1ptCO₂ runs, the diagnosed value of β from CMIP5 models ranges from 0.69 to 0.91 PgC/ppm with a multi-model mean value of 0.80 PgC/ppm. For comparison, the β diagnosed from the NESM v3 simulations is 0.88 PgC/ppm at the end of the simulation. The declining trend in β is found after ~550ppm, later than in Hist+RCP8.5 experiments (~400 ppm), consistent with the results in CMIP5 models. Compared with β, the γ parameter from CMIP5 models has a much larger range, and the value at the end of the simulations ranges from -2.4 to -12.1 PgC/K. The larger spread of γ is associated with the spread of the model-simulated climate change and the dependency of carbon cycle processes on climate change. For comparison, our simulated γ parameter is -7.9 PgC/K.

## 4 Discussion and conclusion

In this study, we evaluated the performance of the NUIST Earth System Model (NESM v3) in simulating the present-day ocean biogeochemical cycle. We also investigated the response of oceanic $CO_2$ uptake to the individual and combined effect of increasing atmospheric $CO_2$ and $CO_2$-induced global warming





under SSP5-8.5 and 1ptCO$_2$ scenarios.

The model simulates reasonably well the large-scale patterns of upper ocean nutrients with high concentrations in the Mid-Eastern Pacific, subarctic Pacific, and the Southern Ocean. The NESM v3 simulated global patterns of upper ocean alkalinity and DIC broadly agree with observations with high

alkalinity concentration in the middle-latitude oceans and high DIC concentration in the high-latitude oceans.

Chlorophyll is reproduced in the model with high concentrations in the high-latitude ocean, but there are noticeable negative biases in the Indian Ocean and maritime continent. The vertically integrated NPP in the model broadly agrees with observational NPP diagnosed by Epply-VGPM and CbPM with high NPP

concentrations in the low-latitude oceans. The integrated global ocean NPP from 2003 to 2014 simulated by the NESM v3 is 45.1 PgC yr$^{-1}$, which is comparable with observation-based estimates and CMIP5 model-simulated results of 30.9-78.7 PgC yr$^{-1}$. Our results suggest that temperature-dependence is necessary to be considered when estimating marine NPP.

The NESM v3 simulates reasonably well the global pattern of sea-air CO2 flux. The model-simulated

cumulative anthropogenic CO$_2$ uptake from the pre-industrial time to the year 2016 is 149 PgC, which compares well with data-based estimates of $150 \pm 20$ PgC. In the 1ptCO$_2$ run, by year 140, the NESM v3 simulated carbon-concentration sensitivity parameter is 0.8 PgC/ppm, and the carbon-climate sensitivity parameter is -7.9 PgC/K, indicating that increasing atmospheric CO$_2$ alone increases oceanic CO$_2$ uptake while global warming alone decreases oceanic CO$_2$ uptake. These estimated sensitivity parameters are

comparable with those estimated by CMIP5 models. The nonlinear of CO$_2$ uptake is mainly simulated in the high-latitude oceans and is associated with equilibrium climate sensitivity (ECS). The ECS in the NESM v3 is about 10% higher than the CMIP5 ensemble (Cao et al., 2018), and thus results in a relatively larger nonlinearity of oceanic CO$_2$ uptake. In this study, the diagnosed nonlinearity accounts for about 10% of the total oceanic CO$_2$ uptake.

The model captures many aspects of the spatial structure of biogeochemical fields and their responses to climate change. However, some defects and their underlying causes should also be emphasized, which are helpful to future model development.

Slight overestimations of nutrients are found in the Pacific and the Southern Ocean (Fig. 1), where the





strong iron limitation is simulated (Fig. 5). The strong iron limitation in these areas limits biological activities, therefore reducing the uptake of nutrients by phytoplankton. In the Indian Ocean, the underestimation of nutrients is associated with the weak upwelling (figures not shown) that suppresses the nutrient entrainment to surface water. The low-level nutrients in the Indian Ocean reduce the biological

activities and then result in negative biases of NPP and chlorophyll. Also, in a relatively coarse resolution model, the negative biases of NPP and chlorophyll in the Indian Ocean could be associated with the poor descriptions of mesoscale and submesoscale processes (McGillicuddy et al., 1998; Lévy et al., 2001b). The latitudinal-depth distribution of nutrients broadly agrees with observations, but the simulated high-concentration centers in the Northern Pacific are too strong and too deep (Fig. 2). The same problem is

also found in the IPSL-CM5A-LR simulated nutrients (figures in the supplement). By decomposing phosphate into its preformed and regenerated components (Fig. 3), then comparing them with the total phosphate distribution, we found that excessive remineralization in the deep ocean is the main cause of the overestimated nutrients in the Northern Pacific.

The negative deviations of alkalinity of ~60 mmol / m$^3$ and DIC of ~40 mmol C / m$^3$ are simulated near

the maritime continent, where the model underestimates the surface salinity of ~ 2 PSU. Cao et al. (2018) found that this underestimation of surface salinity is caused by excessive precipitation in this region. In the high-latitude ocean, the model underestimates SST of about 1°C, indicating stronger convective mixing, which would lead to the overestimation of alkalinity. Similar to the latitudinal-depth distributions of nutrients, the model-simulated high concentration centers of alkalinity and DIC are too strong and too

deep in the Northern Pacific. Excessive remineralization in the deep ocean consumes a large amount of oxygen and releases dissolved organic carbon and nutrients. To better evaluate the NUIST-CSM simulated ocean dynamics and the ocean carbon cycle, the simulation of natural and bomb $^{14}$C will be implemented in future versions of NESM.

It is expected that an improved representation of physical circulation would result in an improved

representation of the marine biogeochemical cycle. Overall, NESM v3 can be  applied in the future to study interactive feedbacks between the ocean carbon cycle and climate change and the underlying mechanisms.  . .

**Code and data availability.**



The source code of NESM v3, together with all input data are saved in one compressed file, which can be downloaded from: https://doi.org/10.5281/zenodo.3524938 after registration. Also, a user guide describing the installation instructions, driver scripts, and software dependencies can be found in the repository at the same link. The simulation results illustrated in this study can be made available upon request to the authors.

**Author contributions.** Yifei Dai and Long Cao performed the simulations, analyzed the experiments, and made the figures. The NUIST ESM team led by Bin Wang provided the code of NESM v3 used in this study and Bin Wang provided helpful discussions. Yifei Dai, Long Cao, and Bin Wang all contributed to the writing of the manuscript.

**Acknowledgement**

Long Cao is supported by the National Natural Science Foundation of China (41675063; 41422503). Bin Wang acknowledges the support by the Nanjing University of Information Science and Technology through funding the joint China-US Atmosphere-Ocean Research Center at the University of Hawaii. Yifei Dai acknowledges the support by the China Scholarship Council by providing a scholarship under the State Scholarship Fund. This is the ESMC publication number XXX and IPRC publication number YYYY.

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



**Table 1.** Global ocean anthropogenic $CO_2$ uptake simulated by NESM v3 during different periods compared against data-based estimate (Ciais et al., 2013) (it is noted that the pre-industrial time in this study represents the year 1850 while it represents 1750 in IPCC AR5).

| | Pre-industrial-2011 Cumulative PgC | 1980-1989 PgC yr$^{-1}$ | 1990-1999 PgC yr$^{-1}$ | 2000-2009 PgC yr$^{-1}$ | 2002-2011 PgC yr$^{-1}$ |
|---|---|---|---|---|---|
| IPCC AR5 | 155±30 | 2.0±0.7 | 2.2±0.7 | 2.3±0.7 | 2.4±0.7 |
| NESM v3 | 137.2 | 1.7 | 2.0 | 2.3 | 2.3 |



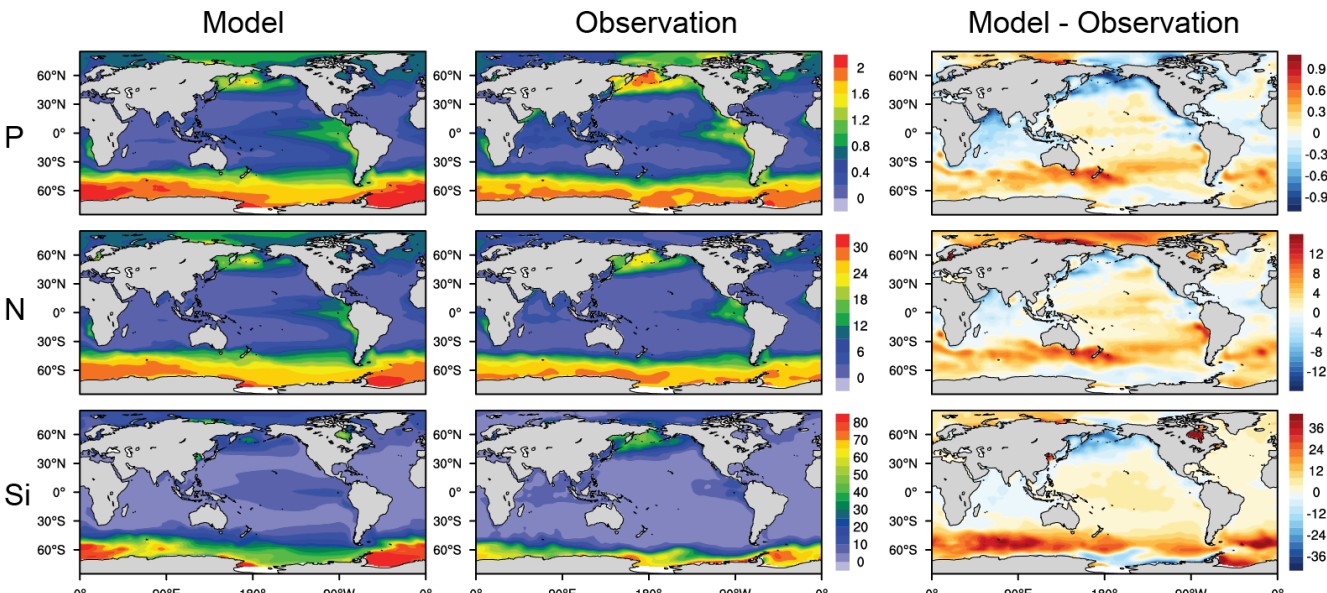

**Figure 1.** Annual mean upper ocean (averaged in the upper 100m) distribution of phosphate ($PO_4^{3-}$), nitrate ($NO_3^-$), and silicate ($SiO_4^{2-}$) from 1985 to 2014 from the NESM v3 simulations (FC) and the WOA18 observation dataset (in unit of mmol/m$^3$).







**Figure 2.** The latitude-depth distribution of silicate (a), phosphate (b), and nitrate (c) averaged from 1985 to 2014 (FC) compared with the WOA18 observation dataset (with a unit of mmol m$^{-3}$). a, b, c represent the silicate, phosphate, and nitrate, respectively. 0 and 1 represent the distributions in the Pacific Ocean, 2 and 3 represent the distributions in the Atlantic Ocean, and 4 and 5 represent the distributions in the Global Ocean.



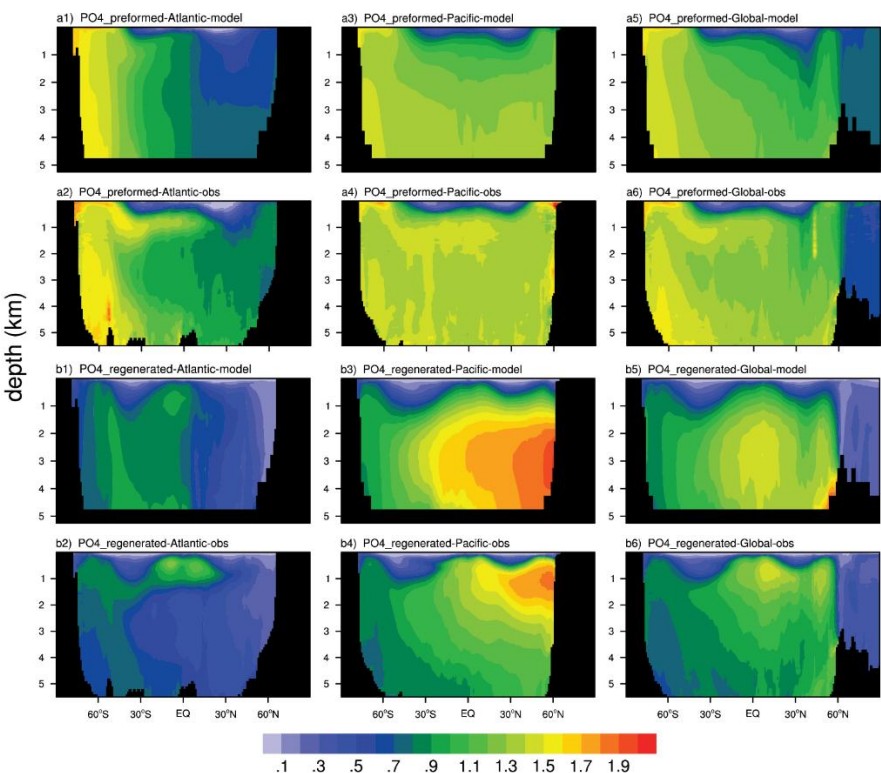

**Figure 3.** The latitude-depth distribution of preformed and regenerated phosphate concentration (mmol m$^{-3}$) simulated by the model and analyzed from WOA18 observation dataset in the Pacific, Atlantic, and global Ocean. The panels from a1 to a6 show the preformed component, while the panels from b1 to b6 show the regenerated component.





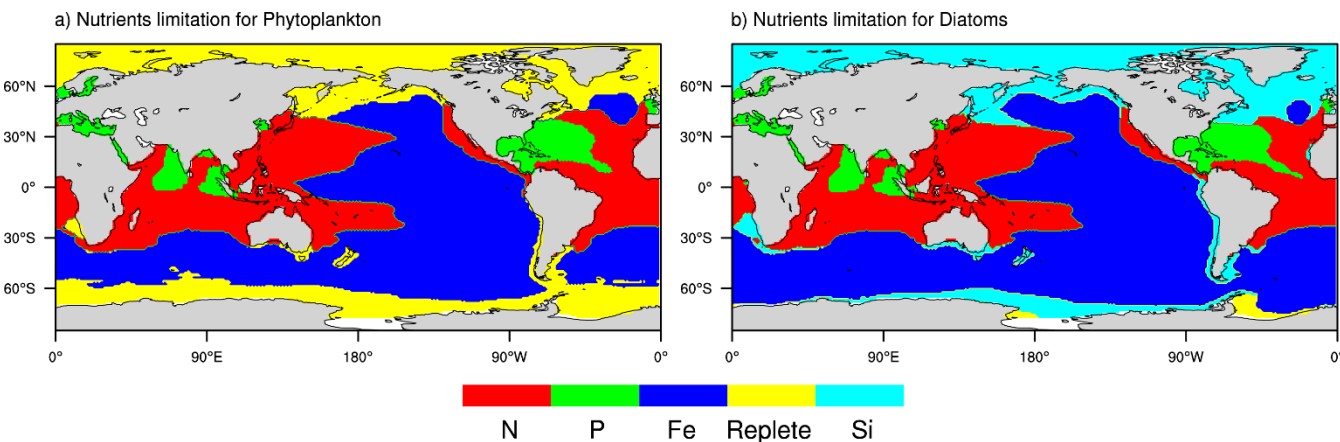

**Figure 4.** Diagnosed pattern of nutrients limitation over the annual time scale for nanophytoplankton and from 1985 to 2014 in the FC simulation. Shade of each color indicate the factor that most limits growth. Replete means nutrient concentrations are sufficient for the phytoplankton growth (growth rate is greater than 90% of their maximal growth rate).



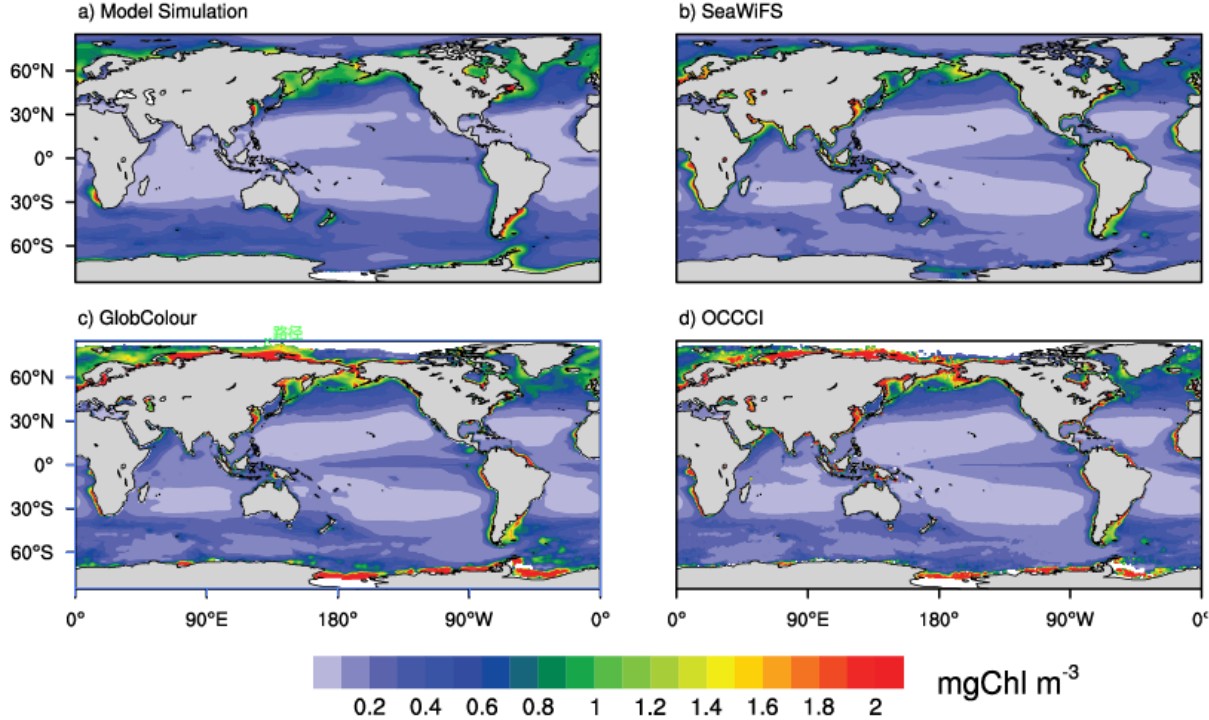

**Figure 5.** Annual mean surface chlorophyll concentration (mg Chl m$^{-3}$) from the NESM v3 FC simulations (a; from 1998 to 2014), the SeaWiFS dataset (b; from 1998 to 2010), the GlobColour merged dataset (c; from 1998 to 2014), and OCCCI merged dataset (d; from 1998 to 2014).



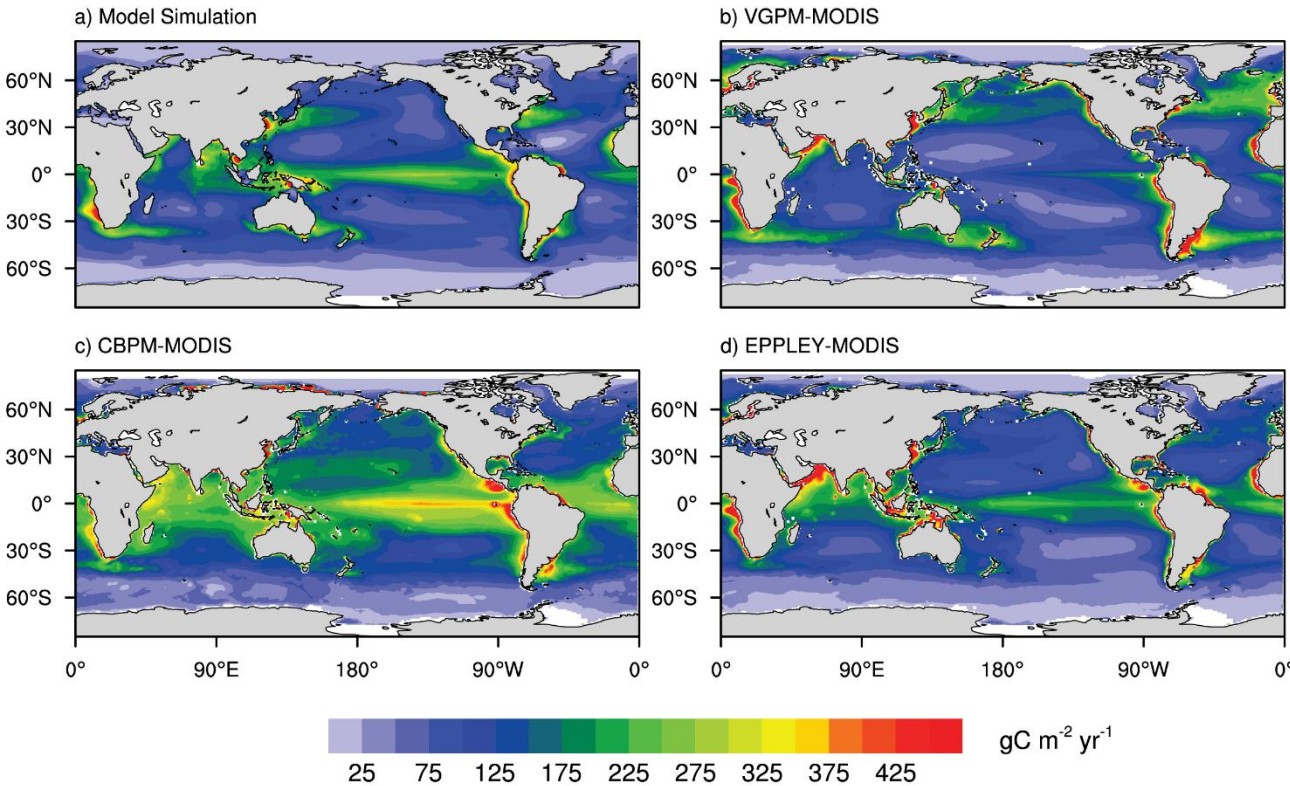

**Figure 6.** Annual mean distribution of vertically integrated net primary production (g C m⁻² yr⁻¹) averaged from 2003 to 2014 from the NESM v3 FC simulations (a) and MODIS observation-based estimates (b: VGPM; c: Epply-VGPM; d: CbPM).





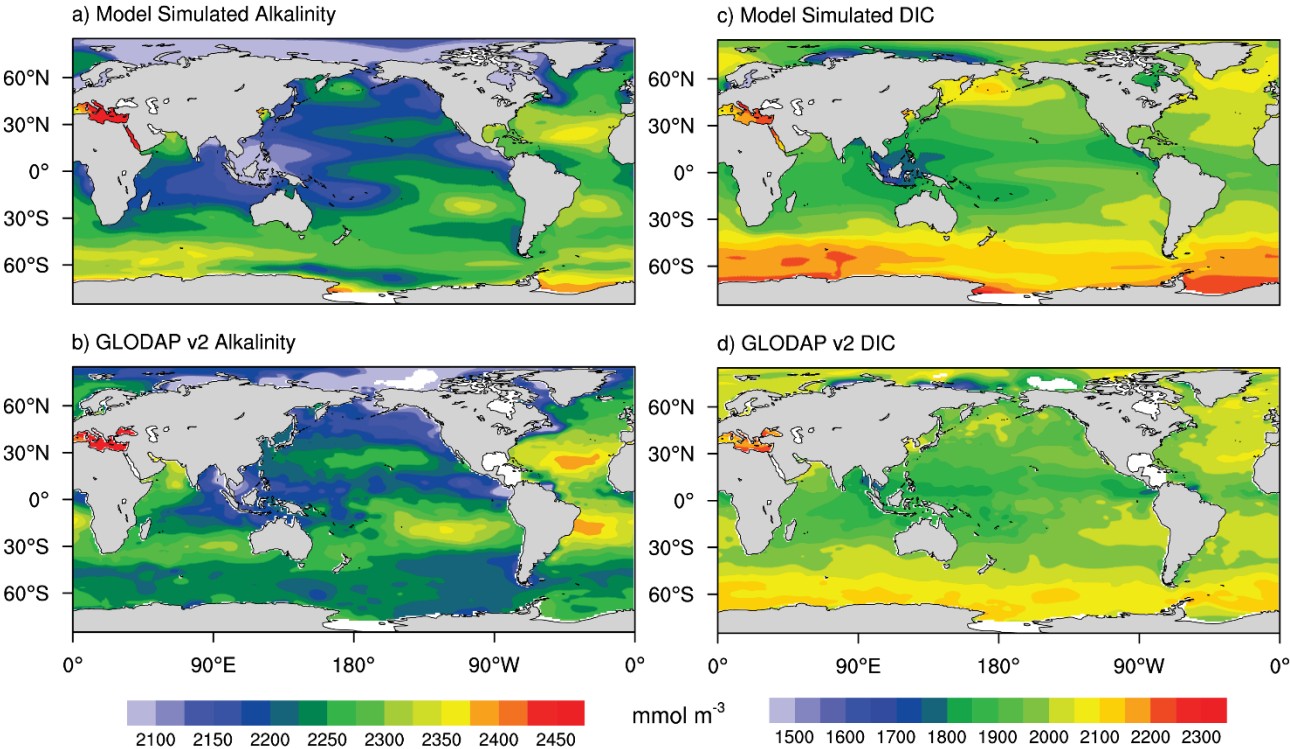

**Figure 7.** Annual mean distributions of upper ocean mean (0-100m) alkalinity (mmol m-$^3$) (a, b) and DIC (mol m-$^3$) (c, d) averaged from1985 to 2014 from the NESM v3 FC simulations (a, c) and GLODAP v2 (b, d).





**Figure 8.** The latitude-depth distributions of the alkalinity (a) and DIC (b) averaged from 1985 to 2014 (FC) compared with GLODAP v2 observations (with a unit of mmol m$^{-3}$; a1, a2, b1, and b2: over the Atlantic Ocean; a3, a4, b3, and b4: over the Pacific Ocean; a5, a6, b5, and b6: over the global ocean).





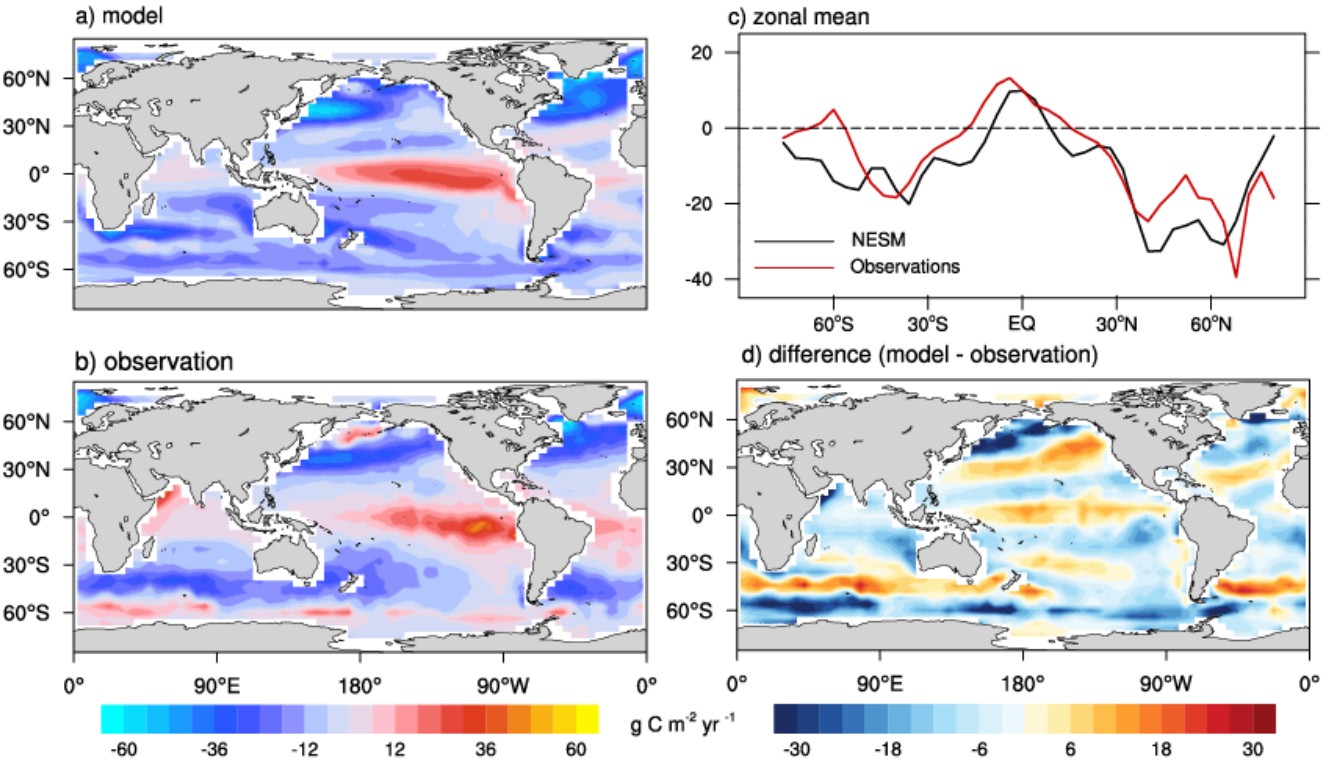

Figure 9. Model-simulated sea-air $CO_2$ flux (g C $m^{-2}$ $yr^{-1}$) in the year 2000 compared with data-based observational estimates (Takahashi.et al., 2009). Spatial distributions of model simulation (a), observation (b), the difference between model and observation (d), and zonal mean pattern of model simulation and observation (c). Positive values represent $CO_2$ flux out of the ocean, and negative values represent $CO_2$ flux into the ocean.



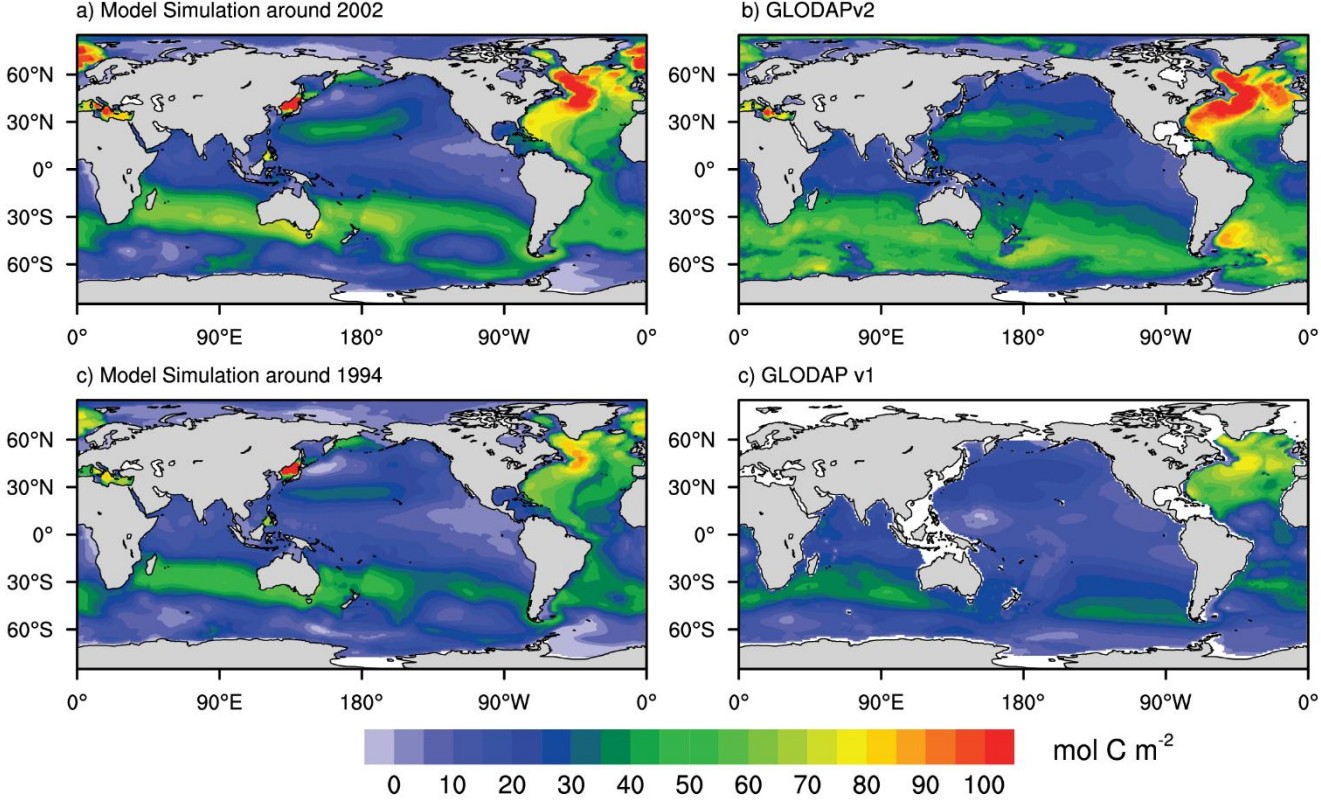

**Figure 10.** Vertically integrated column inventory of anthropogenic DIC (mol C m$^{-2}$) from the FC simulation (a, c) and GLODAP v1 and GLODAP v2 observation (b, d). Model simulation results are averaged from 2000 to 2004 (a) and from 1992 to 1996 (c), while the observation is normalized to the year of 2002 (b) and 1994 (d).



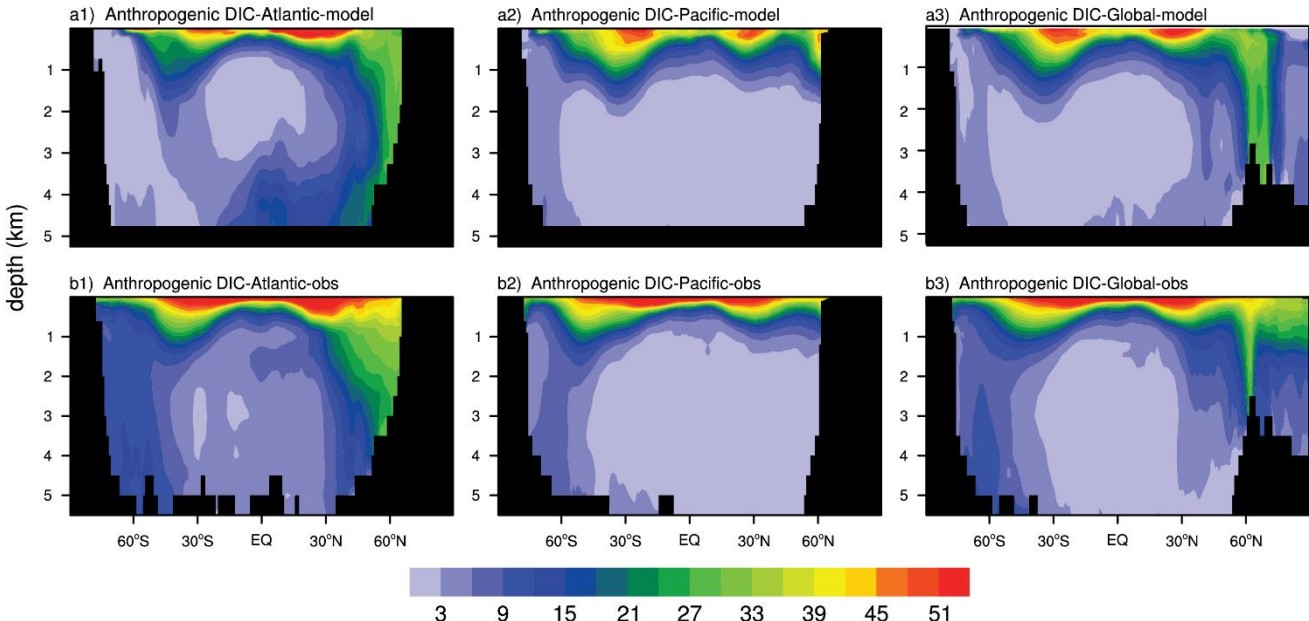

**Figure 11.** Zonal mean latitude-depth distribution of anthropogenic DIC (mmol C m$^{-3}$) distribution from the FC simulation (a1: Atlantic, a2: Pacific, and a3: Global) and data-based estimates (GLODAP v2) (b1: Atlantic, b2: Pacific, and b3: Global).



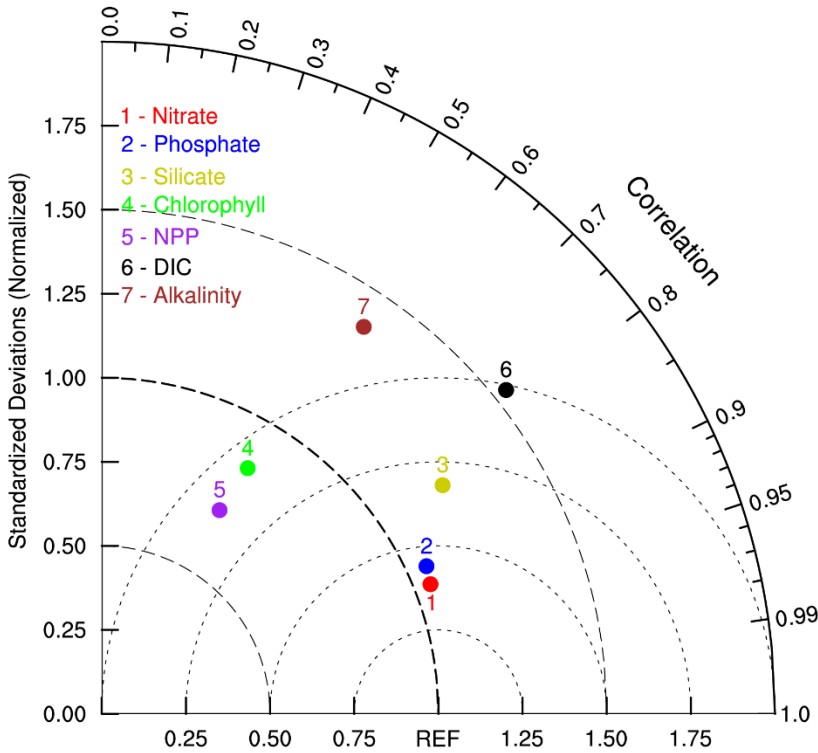

**Figure 12.** Taylor diagram comparing statistical patterns of annual mean carbon-related fields between

the NESM v3 simulation (FC) and corresponding observations, including upper ocean nitrate,

phosphate, silicate, alkalinity, chlorophyll concentration, and vertically-integrated net primary

production. NPP is compared with the CbPM, and chlorophyll is compared with SeaWiFS. All fields are

normalized by the standard deviation of corresponding observations. Thus, observation fields have a

standard deviation of one, which is represented by REF. The distance between the model points and the

reference point indicate the root-mean-square (RMS) difference between model simulation and

observations.





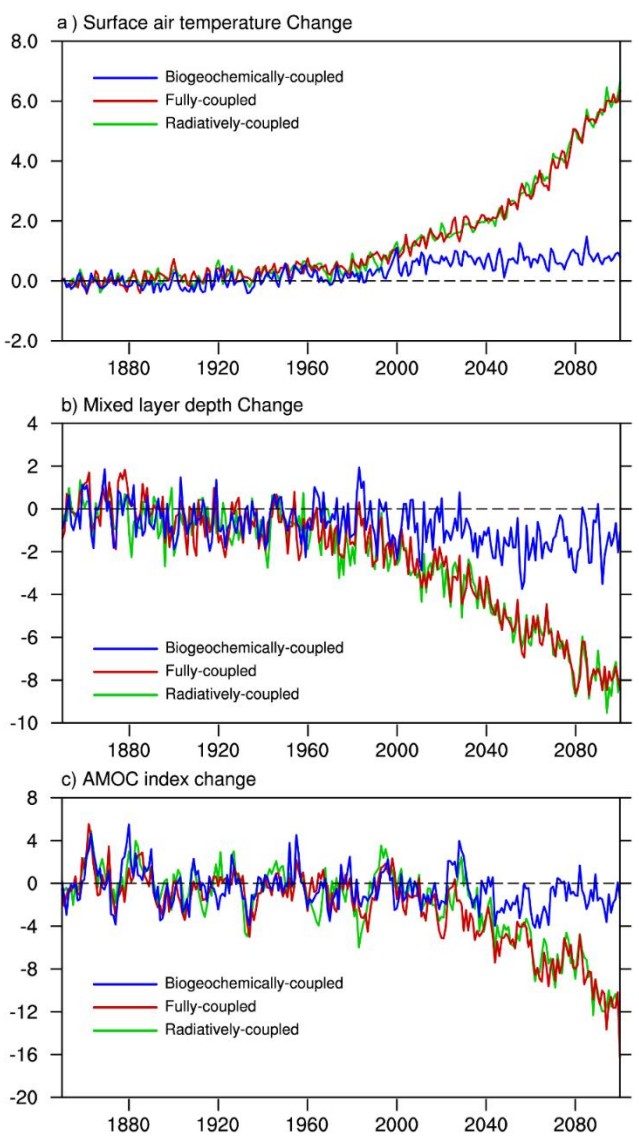

**Figure 13.** Time series of climate changes (minus control simulation) from 1850 to 2100 for the simulation of fully-coupled, biogeochemically-coupled, and radiatively-coupled simulations. (a) global and annual mean surface air temperature, (b) global and annual mean mixed layer depth (the depth where the difference in potential density is 0.01 kg m$^{-3}$ relative to the sea surface) and (c) Atlantic meridional overturning circulation index (maximum zonal mean stream function in the Atlantic Ocean at 30°N).





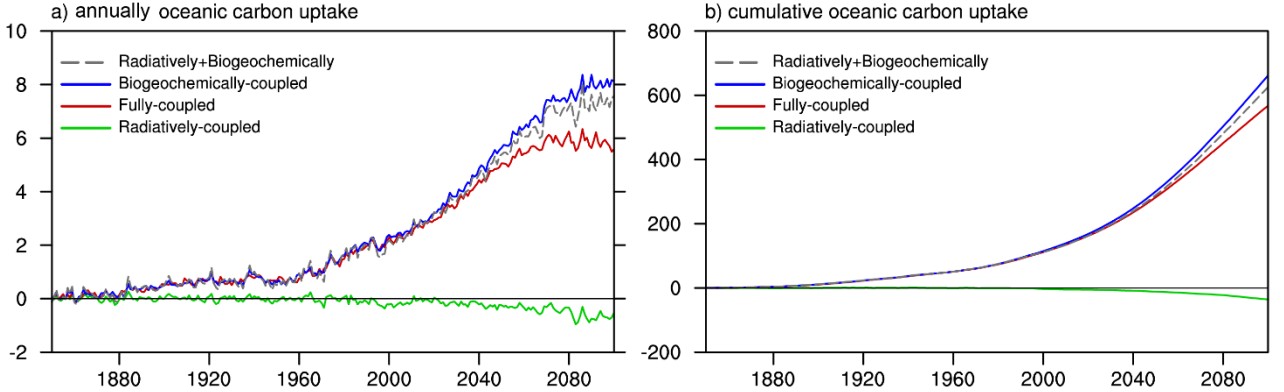

**Figure 14.** The NESM v3 simulated (a) annually oceanic $CO_2$ uptake change (minus control simulation) and (b) cumulative oceanic $CO_2$ uptake for the simulations RC, BC, FC, and the linear sum of BC and RC from 1850 to 2100 (in unit of Pg C).





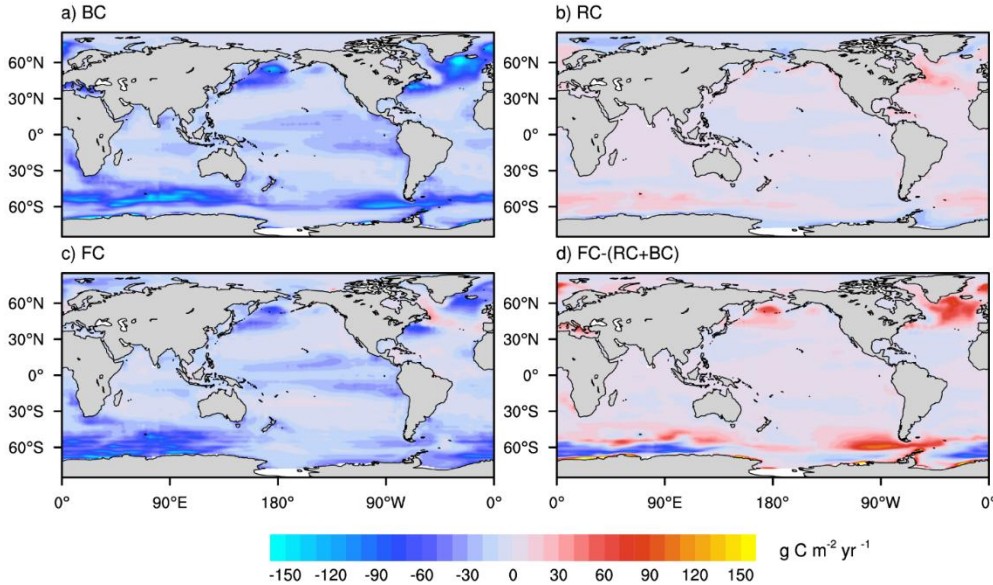

**Figure 15.** Spatial distribution of anthropogenic sea-air $CO_2$ flux at the end of the 21st century (mean of 2091-2100 minus control simulation) from the (a) BC, (b) RC, and (c) FC, respectively. Also shown is the difference between FC simulation and the sum of RC and BC simulations (FC-RC-BC). Positive values represent $CO_2$ flux out of ocean, and negative values represent $CO_2$ flux into the ocean.



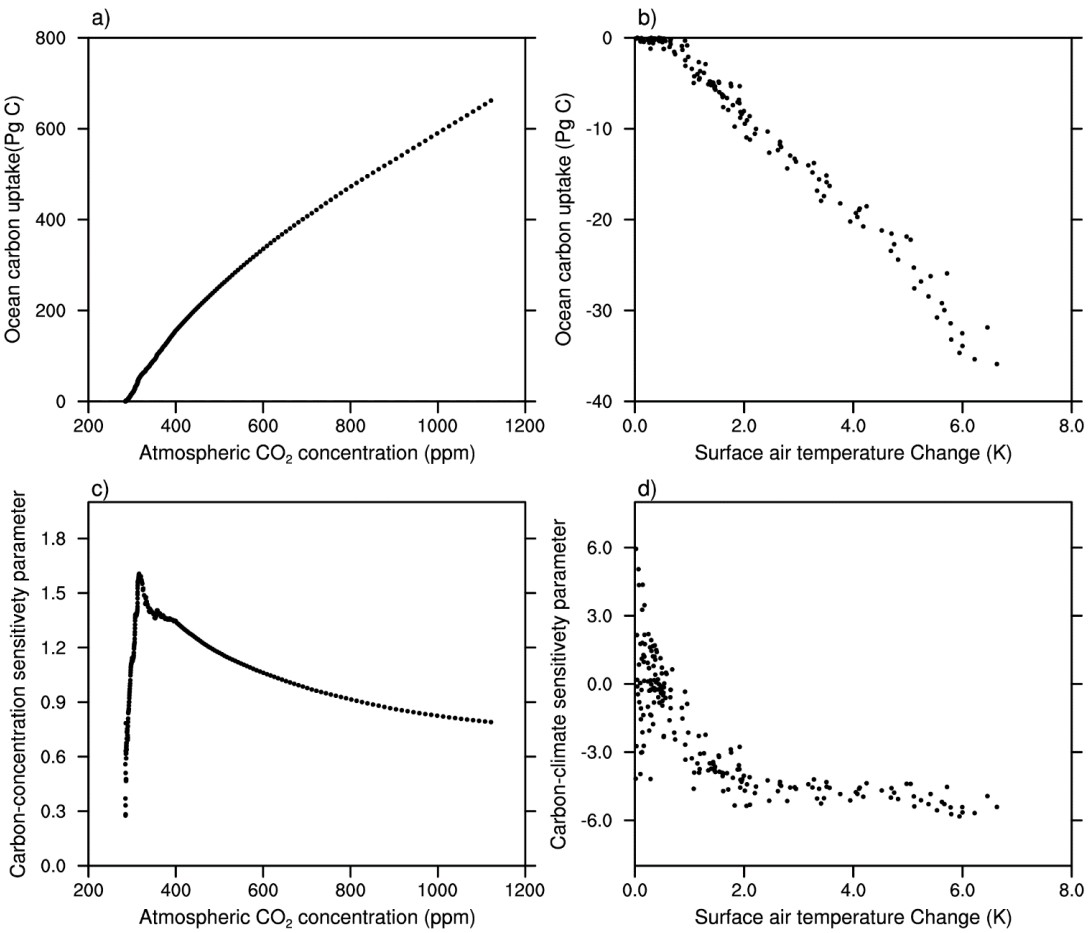

**Figure 16.** The cumulated oceanic $CO_2$ uptake against (a) the atmospheric $CO_2$ in the BC simulation and (b) the global mean surface air temperature change in the RC simulation. Also shown is time evolution of diagnosed carbon-concentration sensitivity parameter as a function of atmospheric $CO_2$ (c) and carbon-climate sensitivity parameter as a function of global mean surface air temperature change (d).



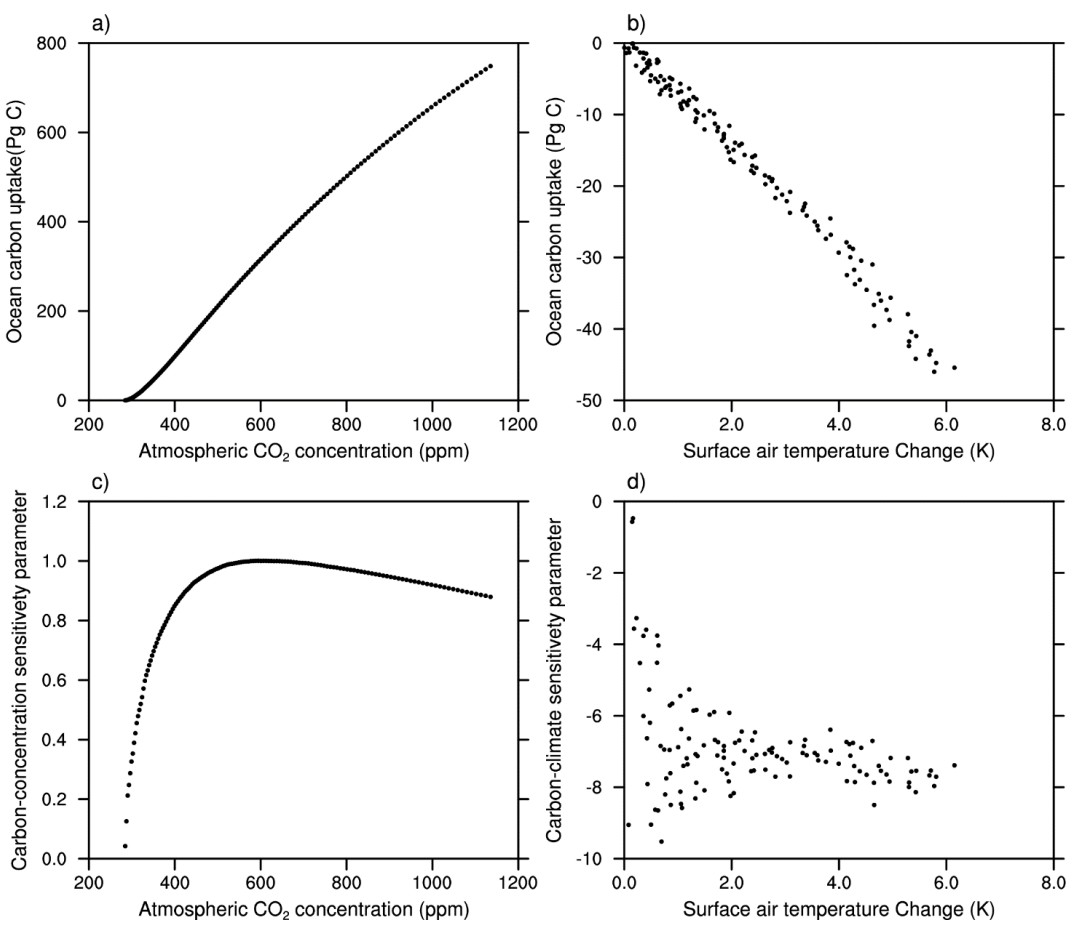

**Figure 17.** Same as Figure 16, but for the 1pt CO₂ runs.