# Peer review of "Marine biogeochemical cycling and oceanic CO2 uptake simulated by the NUIST Earth System Model version 3 (NESM v3)"

_Geoscientific Model Development, 2019_

## Short Comment (SC1) · 4 Dec 2019

Dear authors,

please add the definition of NUIST in the paper. This abbreviation is not proper introduced in the article. Additionally, I suggest to mention the acronym NESM v3 also in the title.

Best regards,

Astrid Kerkweg (GMD executive Editor)

---

## Author Comment (AC1) · 5 Dec 2019

Dear Astrid Kerkweg,

Thank you very much for your comments. The acronym NUIST stands for Nanjing University of Information Science and Technology. We will add the definition in the abstract. The title will be changed to 'Marine biogeochemical cycling and oceanic CO2 uptake simulated by the NUIST Earth System Model version 3 (NESM v3)'. All these revisions will be done after the interactive discussion.

Best,

Yifei Dai

---

## Referee Comment (RC1) · Anonymous Referee #1 · 10 Dec 2019

Review of "Marine biogeochemical cycling and oceanic CO2 uptake simulated by the NUIST Earth System Model version 3" by Dai et al.

General comments

This paper provides a description and evaluation of marine biogeochemical simulation of the NESM v3. Given that this model is new to the CMIP community (participating in CMIP6), it is important to provide such description and evaluation paper to discuss the strengths and limitations of the model to help the end users of CMIP6 archives. Below I provide a few general comments, followed by specific comments and editorial corrections, to help improve the presentation of the paper. I believe that the authors

can address these concerns.

The motivation of this paper is not clearly stated in the introduction section. I assume it is to evaluate the model performance of marine biogeochemical fields (e.g., P1, L10 and P3, L23), but the authors should describe why doing so is important for this particular model. I think this could be clearly stated by having a paragraph starting with a sentence like "The objective of this paper is to . . ." in the second last paragraph of the Introduction section, and by linking to CMIP6.

Restructure. I find that the flow of the paper could be improved if all the method stuff goes into the Method section. See my specific comments.

Concluding paragraph. The very last paragraph of the Discussion and Conclusion section is very weak (P25, L24). Also the two dots ".." at the very end indicates that it is unfinished. Please work on this last paragraph to provide the take-home message of the manuscript.

Model minus observation figures. Some figures can be improved by having a model-minus-observation difference subplots. This is done for Figures 1 and 9, but not for the others. Doing so makes it easy to show where the model has positive and negative biases and by how much, and helps understand the text (for example, it is hard to notice the negative biases mentioned in P24, L7 just by looking at Fig. 5).

Specific comments

Title. I think the acronym NESM v3 should be mentioned in the title if that's the preferred acronym for CMIP6 exercise.

Acronyms. All acronyms should be defined for the first time they appear in the text. For example, in the abstract, CO2, CMIP5, 1ptCO2. I suggest the authors to check throughout the manuscript.

P3, L8 to L24. I feel that the details of NESM v3 mentioned in these two paragraphs belong to the method section. Instead, provide the objective paragraph here with a very

brief introduction of NESM v3. At this point, the readers are not interested in knowing the details of NESM v3, but rather they want to know why NESM v3 is important and deserves an evaluation paper.

P5, L1. This sentence sounds to me that salinity of 4 PSU is added into the ocean when ice melting, but is this correct? I think the sea-ice salinity is fixed at 4PSU which is used to calculate the ice-ocean salt flux, which should actually result in dilution of seawater during ice melting (unless the ocean salinity is less than 4 PSU).

Sec. 2.2. Perhaps not so important for the interpretation of results, but to complete the description, mention the initial and boundary conditions for biogeochemical state variables. Were they initialized to the WOA18, GLODAP v2? Was the river discharge of biogeochemical state variables prescribed?

P7, L9 and L10. Briefly describe what it means by "offline".

P7, L22. Why follow the protocol of CMIP5 for 1ptCO2 and not that of CMIP6 (one of the DECK experiments)?

P8, L16. Consider moving this paragraph earlier (at the beginning of the section) to give a broad picture up front.

P8, L24. WOA18 gives nutrient concentrations in units of umol/kg. Briefly mention how they were converted to mmol m-3 for model-obs comparison (e.g. Fig.1).

P9, L1. Briefly mention how these products compare or differ. I mean, do they not all incorporate SeaWiFS? Also, GlobColour and OCCCI are both merged products (and they look pretty much the same; Fig. 5), so the readers may be curious to know why these two similar products deserve comparison.

P9, L21. Suggest to remove this paragraph as it was already mentioned in P8, L20.

P10, L4. In addition to these physical processes, iron limitation is another main reason for high macronutrient levels in this region, which should be mentioned here.

P10, L10. Refer to the figure (Fig. 1 c,f,i) at the end of the first sentence.

P10, L22 to P11, L4. This paragraph belongs to the methods section.

P11, L2 to P12 L3. Again this paragraph belongs to the method section. Also, mention what the half-saturation constant is to set to in the model for each nutrient.

P12, L21. I don't think the chlorophyll levels are high for the equatorial Pacific and the Southern Ocean. They may be "relatively" high compared to the surrounding seas, but the absolute magnitudes are low.

P13, L7. I am not sure what it means by "the intermediate concentration regions". Maybe provide a number?

P13, L11. Instead of the International Date Line, refer to the longitude coordinate? Not every reader knows the exact location of the date line. This and the next sentence can be easily identified if the model-obs subplots are provided.

P13, L14. From 1998? Fig. 6 says from 2003.

P13, L25. PAR should be defined earlier in the text, and here just write as PAR.

Figure 6. Caption and the figure text do not match. Please check more carefully. In the caption, b = VGPM, c = Eppley-VGPM, d = CbPM, whereas in the figure, b = VGPM-MODIS, c = CbPM-MODIS, d = Eppley-MODIS.

P14, L25. Here and elsewhere, the term "deviation" is used to refer to the model-obs difference. This is a bit confusing because standard deviations are also used in the later analysis (e.g. P16, L2). Perhaps, use "difference" instead of "deviation".

P15, L10. Again, this is where having the model-obs difference subplot would be helpful to support this sentence.

P16, L9. "due to the 3-dimensional correction . . .", unclear what this means. Add one or two sentences to explain.

P16, L12. Provide a reference for the observation value quoted here.

Table 1. As noted in the caption, the pre-industrial years between NESM v3 and IPCC AR5 differ by 100 years. Does this explain why the IPCC AR5 value is higher than NESM v3 because the former incorporates additional 100 years for the cumulative quantity? Also, in the caption, describe what the plus/minus values represent.

P17, L25. Should this paragraph have its own section? It is beyond oceanic co2 uptake (Sec 3.4).

P18, L21. ", which is associated ... deep ocean," this middle block of sentence is not supported by any figure or reference, and also is unnecessary. It can just be removed and simply be stated as "The reduction of mixed layer depth indicates a more stratified upper ocean ...".

Sec 3.5.1. This section only discusses the results of the FC simulation. Maybe provide some comments on the different simulations (e.g. BC vs RC simulations).

P19, L11. This first paragraph is already mentioned in the previous section, so why repeat here?

P21, L21 to P22, L8. This block of paragraphs belongs to the Method section. Having this much of methodological details in the results section breaks the flow of the paper. Please consider moving it to the Method section.

P22, L16. "Therefore, ..." this sentence is unclear to me. Especially for the carbon-climate parameter, which can be both positive and negative as shown in Figure 16d. Do the authors mean that it is negative in the year 2100?

P23, L3. This paragraph could move into the Method section and be combined with the block of paragraphs describing the sensitivity parameter derivation. Also, "4xCO2" is unclear.

P24, L12. "Our results suggest ..." How does your result support that temperaturedependence is necessary?

P25, L1. "The strong ...", but why does NESM v3 overestimate nutrients? The iron limitation is too strong? Too strong vertical mixing? A few speculations can be helpful.

P25, L10. Why do the results of IPSL-CM5A-LR appear here and not in the results section? Also, why choose this particular CMIP ensemble member over others? I don't really see the point of adding the comparison with IPSL-CM5A-LR.

Editorial corrections

P1, L15. "total CO2 uptake" –> Use subscript for "2"

P3, L2. "; 2)" –> "; and 2)"

P3, L4. "the effect of CO2 concentration" –> "the effect of increasing atmospheric CO2 concentration"

P4, L7. Should "NUIST-CSM-2.0.1" be "NESM v3"?

P4, L16 (and elsewhere). Add a space between 10 and m (10 m instead of 10m).

P5, L18-19. ": nanophytoplankton and diatoms," –> "(nanophytoplankton and diatoms)" and similarly for zooplankton.

P5, L22. "photosynthetic" –> "photosynthetically"

P6, L9. POM is already defined in P5, L25.

P6, L9. "diatoms silicate", should this be "biogenic silica"?

P6, L9. "described by ... corresponding to" –> "is partitioned into"

P12, L2. "when all nutrients ... than 0.9." –> "when the annual mean nutrient coefficients are greater than 0.9 for all nutrients."

P12, L14. SeaWiFS should be defined in P9, L1. OCCCI was already defined in P9, L2.

P12, L19. "plankton" –> "phytoplankton"

P14, L23 Add a space between alkalinity and are.

Fig. 7. Superscript for alkalinity units. Also add a space between from and 1985.

P14, L23. Add a space between alkalinity and are.

P15, L9. Add a space between ocean and means.

P16, L2. I don't think SD has been defined previously. If so, define here.

Figure 10. In the figure, the last subplot is labelled as "c) GLODAP v1", which should be "d) GLODAP v1".

P16, L24. "receptively" –> "respectively".

Figure 12. Caption: "statistical patterns" –> "spatial patterns"(?); "carbon-related" –> "biogeochemical"; "upper ocean" –> "upper 100-m ocean". Provide the information on observations for nutrients, DIC, and alkalinity, such as done for NPP and chlorophyll.

P18, L3. "to atmospheric" –>"to increasing atmospheric".

P18, L5-L8. "presented" –> "present" and "compared" –> "compare".

P18, L20. "and acting to mitigate" –> "and".

P18, L21. "MLD is seen decreasing" –> "Modeled MLD decreases".

P18, L24. "is seen in" –> "is projected for".

Figure 15. In the caption "FC-RC-BC" –> "FC-RC+BC"?

Figure 16. Caption "cumulated" –> "cumulative". "atmospheric co2 (c) and . . ." –> "(c) atmospheric co2 and (d) . . ."

P21, L25. ". The" –> ", the". Similarly for P22, L4.

P23, L11. "estimated by CMIP5 models range" –> "estimated for CMIP5 models ranging"

P24, L3. "Mid-Eastern" –> "Eastern"

P24, L14. The numbers quoted in this paragraph is inconsistent with the ones appeared in the Results section. Specifically, year 2016 (should this be 2011? Table 1), 149, 150 +/- 20, and 0.8. Please check these numbers with the Results section.

P24, L20. "nonlinear of " –> "nonlinear response of oceanic" P25, L16. "precipitation in" –> "precipitation simulated in"

P25, L18. "which would lead" –> "which leads"

P25, L21. "NUIST-CSM", should this be "NESM v3"?

---

## Referee Comment (RC2) · Anonymous Referee #2 · 18 Mar 2020

Marine biogeochemical cycling and oceanic CO2 uptake simulated by the NUIST Earth System Model version 3

by Yifei Dai, Long Cao, Bin Wang

1) General comments

Dai and co-authors evaluate the ability of their earth system model NESM v3 to represent the carbon cycle (and, particularly, the CO2 uptake) and the representation of several marine biogeochemical tracers (nutrients, alkalinity, DIC, chlorophyll and net primary production). NESM v3 performances are compared with observations and, occasionally, to CMIP5 models. As regional discrepancies are identified, the authors

discussed their physical (e.g. weak upwelling in the Indian Ocean, strong convective mixing at high latitudes and more generally, shortcomings in simulated ocean circulation) or biogeochemical (iron limitation in the Southern Ocean, excessive remineralization in the deep Northern Pacific) origins.

2) Relevance of the subject

Such a paper evaluating the limits of a modelling platform can be very useful to the scientific community which is going to use and analyse NESM v3 outputs, especially if the model has contributed to the CMIP6 Intercomparison Project.

However I did not really understand from the text if the model described and used in this paper has really been a part of CMIP6: p.3, l.10: "as a registered model of CMIP6" but p.7, l.21: "following the protocol of CMIP5" Maybe the authors may explain why not using the protocol of CMIP6 (to be CMIP6 fully compliant) ?

3) General structure

The readability could be improved by a better structure. Please have a look at Séférian et al. (2019, https://doi.org/10.1029/2019MS001791) which provides an evaluation of CNRM earth system model for CMIP6 by comparing it to observations, as well as to CMIP5 multi-model ensemble, and to an earlier version of the same model.

Keeping the introduction in its actual state (i.e. focusing on carbon uptake), I suggest to move the description of NESM v3 found in the introduction (p.3, l11-16) to the dedicated section (2.1.1). But I would rather expect a more focused introduction, relaying previous/other model evaluations of the carbon cycle and uptake. As it is, the scope of the introduction is a bit too wide.

4) Results

I suggest to discuss the magnitude of the nutrient biases obtained in section 3.1 in regard of those obtained with other models like CMIP5 models (maybe a short summary of the published CMIP5 literature on these aspect will be enough). This would help

the reader to know how NESM v3 places itself in the CMIP models diversity. This is also true for the section relative to the Taylor diagram (Fig. 12): please see my specific comment.

The section 3.5.2 discussing the coupling between the "radiative" (i.e. in this case only atmospheric radiation is affected by changing concentrations of atmospheric $CO_2$) and "biogeochemical" (i.e. in this case only the ocean carbon cycle is affected by changing atmospheric $CO_2$) sensitivity experiments is quite interesting. However if the motivation of this paper is to evaluate model skills in modeling carbon-related biogeochemical species, the study of the non-linearity of their sum appears a bit beyond the lines of the paper. I would recommend that either the authors restructure (a bit) the current draft or clarify the aim of their study.

5) Discussion

I would expect of a paper aiming at evaluating a model that the "Discussion and Conclusion section" would give more details of how this model behaves (in terms of modeled carbon cycle and $CO_2$ uptake here) in comparison with other models or in the context of the other CMIP models. If such comparisons are occasionally done in the current draft version, it would be valuable to systematise them.

6) Language

I would recommend a careful reading which may easily help to correct the typing errors.

7) Specific comments

p.3, l.19: modes -> models

p.4, l.4: includes -> that includes or including

p.4, l.15: tripole -> tripolar grid

p.7-8, l.27, l.1-4: "To separate the effect of atmospheric $CO_2$ and global warming on the ocean carbon cycle, we performed three types of experiments (biogeochemically

coupled, radiatively coupled, and fully coupled). These types of simulations were also performed by previous studies that investigated the effect of $CO_2$ and global warming on the global carbon cycle (Friedlingstein et al., 2006; Arora et al., 2013; Schwinger et al., 2014)."

I suggest to slightly reorganize the above paragraph in order to properly introduce the list items that follows. I suggest something like that:

"Following Friedlingstein et al., 2006; Arora et al., 2013; Schwinger et al., 2014, we performed three types of experiments (biogeochemically coupled, radiatively coupled, and fully coupled) to separate the effect of atmospheric $CO_2$ and global warming on the ocean carbon cycle: 1) Biogeochemically coupled (BC)...."

p.9, l.18: the modeled result -> the modeled sea-air $CO_2$ fluxes to a $4°x5°$ grid.

p.10, l.3: Fig. 11 -> Fig. 1 ?

p.11, l.21-22: why is the ocean circulation so different in IPSL-CM5A-LR and NESM, as both models share the same oceanic model (NEMO) ?

p.14, l.23: alkalinityare -> alklinity are

p.16, l.11: "," -> "."

p.16, l.24: receptively -> respectively

p.17, l.4: stimulated -> simulated ?

p.17, l.23: Similar to the vertically integrated inventory (Fig. 10): I suggest to add the figure you are referring to, in order to facilitate the reading.

I also suggest to clarify (in the text or, at least, in the caption of Fig. 11) the period on which DIC has been averaged for computing these vertical sections: I guess that these vertical sections have been averaged between 1985 and 2014 as other vertical sections ? As it gives an indication of the temporal persistence of the bias, I would suggest to

mention it. The vertically integrated inventory of Fig. 10 are not representative of the whole period but represent only few years around 2002 or 1994.

p.17, l.25 : it would be very helpful to add other models data or even CMIP5 ensemble mean on this Taylor diagram for all the biogeochemical fields in order to get an idea of how NESM v3 behaves in the current modeling landscape.

p.18, l.20: MLD is seen decreasing -> "MLD is seen to decrease" sounds to me more correct.

p.21, l.3: "some regions of the Northern Atlantic even appear CO2 outgassing". The formulation sounds weird to me, please check its grammatical correctness.

p.25, l.27: ". . ." -> "."

p.26, l.15-16: please replace XXX and YYYY by providing publication numbers, or delete the sentence.

8) Figures

p.40, Figure 2: to improve the readability of this figure, I suggest to separate the 3 oceanic regions (Atlantic, Pacific, Global) and to increase the tick labels and titles. You can keep all the 3x6 subplots on the same figure, but at least try to increase the margin below the 6 subplots related to the Atlantic ocean, and also increase the margin below the 6 subplots related to the Pacific ocean following a pattern like:

a0 b0. c0

a1. b1. c1

<- increased margin

a2. b2 c2

a3. b3. c3

<- increased margin

a4. b4. c4

a5 b5. c5

p.41, Figure 3: same recommendation than for Figure 2, please add an increased margin between the cluster of subplots for preformed PO4 and the cluster of subplots showing regenerated PO4. Increase all ticks labels and titles.

As the whole analysis of Fig. 3 is based on biases values, it would be very helpful to show these biases on Fig. 3. You could show a first line of vertical sections relative to observations and then a second line with biases (model-obs) rather than model means.

p.42, Figure 4: please mention the source of these nutrients limitation patterns. Are they diagnosed from NESM v3 model ?

P.44, Figure 6: be careful with the mismatch between subplots titles and their name in the legend.

p.46, Figure 8: see recommendations for Figures 2 and 3.

p. 51 and 53, Figures 13 and 15: please add units of analysed fields.

9) Bibliography

Séférian, R., Nabat, P., Michou, M., Saint-Martin, D., Voldoire, A., Colin, J., et al (2019). Evaluation of CNRM Earth-System model, CNRM-ESM2-1: role of Earth system processes in present-day and future climate. Journal of Advances in Modeling Earth Systems, 11. https://doi.org/10.1029/2019MS001791

---

## Author Response (AR1)

**Response to reviewers**

We would like to thank the reviewers for the in-depth review and constructive comments. Below we provide point-to-point responses to each comment. Reviewer comments are given in italic and responses are given in bold. Manuscript with revisions marked using track changes is also provided.

**Comments to reviewer1**

*General comments.*

1. *This paper provides a description and evaluation of marine biogeochemical simulation of the NESM v3. Given that this model is new to the CMIP community (participating in CMIP6), it is important to provide such description and evaluation paper to discuss the strengths and limitations of the model to help the end users of CMIP6 archives. Below I provide a few general comments, followed by specific comments and editorial corrections, to help improve the presentation of the paper. I believe that the authors can address these concerns.*

**Response: Thank you so much for your in-depth review and constructive comments. The following are the point-to-point responses and manuscript with revisions marked using track changes.**

2. *The motivation of this paper is not clearly stated in the introduction section. I assume it is to evaluate the model performance of marine biogeochemical fields (e.g., P1, L10 and P3, L23), but the authors should describe why doing so is important for this particular model. I think this could be clearly stated by having a paragraph starting with a sentence like "The objective of this paper is to : : :" in the second last paragraph of the Introduction section, and by linking to CMIP6.*

**Response: Thanks for the comment. We added a paragraph starting with "The objective of this manuscript is …" in the introduction section (P3, L11).**

3. *Restructure. I find that the flow of the paper could be improved if all the method stuff goes into the Method section. See my specific comments.*

**Response: Thank you for the suggestion. We give the analysis method section in the revised manuscript (Section 2.4, P9, L20).**

4. *Concluding paragraph. The very last paragraph of the Discussion and Conclusion section is very weak (P25, L24). Also the two dots ".." at the very end indicates that it is unfinished. Please work on this last paragraph to provide the take-home message of the manuscript.*

**Response: Thanks for the comment. The last paragraph of Discussion and Conclusion Section is rewritten (P26, L19).**

5. *Model minus observation figures. Some figures can be improved by having a model-minus-observation difference subplots. This is done for Figures 1 and 9, but not for the others. Doing so makes it easy to show where the model has positive and negative biases and by how much, and helps understand the text (for example, it is hard to notice the negative biases mentioned in P24, L7 just by looking at Fig. 5).*

**Response: Thanks for the suggestion. We add additional model-observation subplots for figure 3, 5, 6, 7.**

*Specific comments*

1. *Title. I think the acronym NESM v3 should be mentioned in the title if that's the preferred acronym for CMIP6 exercise.*

**Response: Thanks for your comment. The title is changed to 'Marine biogeochemical cycling and oceanic CO2 uptake simulated by the NUIST Earth System Model version 3 (NESM v3)'.**

2. *Acronyms. All acronyms should be defined for the first time they appear in the text. For example, in the abstract, CO2, CMIP5, 1ptCO2. I suggest the authors to check throughout the manuscript.*

**Response: Thanks for your comment. We checked throughout the manuscript.**

3. *P3, L8 to L24. I feel that the details of NESM v3 mentioned in these two paragraphs belong to the method section. Instead, provide the objective paragraph here with a very brief introduction of NESM v3. At this point, the readers are not interested in knowing the details of NESM v3, but rather they want to know why NESM v3 is important and deserves an evaluation paper.*

**Response: Thanks for the comment. We move these two paragraphs to model description section and add one paragraph to clarify the motivation of this paper (P3, L11).**

4. *P5, L1. This sentence sounds to me that salinity of 4 PSU is added into the ocean when ice melting, but is this correct? I think the sea-ice salinity is fixed at 4PSU which is used to calculate the ice-ocean salt flux, which should actually result in dilution of seawater during ice melting (unless the ocean salinity is less than 4 PSU).*

**Response: Thanks for the comment. The sentence is rewritten. Salinity of 4 PSU (salt flux) together with the large amount of fresh water (fresh water flux) is added into ocean when ice melting. Therefore, ice melting results in dilution of seawater, and ice formation results in the increase of salinity (P4, L25).**

5. *Sec. 2.2. Perhaps not so important for the interpretation of results, but to complete the description, mention the initial and boundary conditions for biogeochemical state variables. Were they initialized to the WOA18, GLODAP v2? Was the river discharge of biogeochemical state variables prescribed?*

**Response: Thanks for the comment. The description of initial and boundary conditions for biogeochemical state variables is expanded. The river discharge is prescribed (P6, L18-27).**

6. *P7, L9 and L10. Briefly describe what it means by "offline".*

**Response: Thanks for the comment. The offline simulation means uncoupled run. We use each component model's PI control run results as the initial conditions for the couple model. It is described in the manuscript (P7, L13-16).**

7. *P7, L22. Why follow the protocol of CMIP5 for 1ptCO2 and not that of CMIP6 (one of the DECK experiments)?*

**Response: In this manuscript, 1ptCO$_2$ experiment are used to diagnose carbon-climate and carbon-concentration sensitive parameters. The only differences between the protocol of CMIP5 and CMIP6 for 1ptCO2 experiment, is that CMIP5 only requires 140 years integration, while CMIP6 requires more than 150 years. Considering that we primarily compared 1ptCO$_2$ experiments with CMIP5 model results, we only conduct 1ptCO$_2$ for 140 years, e.g., following CMIP5 protocol.**

8. *P8, L16. Consider moving this paragraph earlier (at the beginning of the section) to give a broad picture up front.*

**Response: Thanks for your suggestion. We moved this paragraph to the beginning of this section (P7, L5-10).**

9. *P8, L24. WOA18 gives nutrient concentrations in units of umol/kg. Briefly mention how they were converted to mmol m-3 for model-obs comparison (e.g. Fig.1).*

**Response: Thanks for your comments. We assume the density in the model and observations are same. Then the unit is converted from umol/kg to mmol/m$^3$ by multiplying modeled density (in unit of kg/m$^3$) (P8, L21-23).**

10. *P9, L1. Briefly mention how these products compare or differ. I mean, do they not all incorporate SeaWiFS? Also, GlobColour and OCCCI are both merged products (and they look pretty much the same; Fig. 5), so the readers may be curious to know why these two similar products deserve comparison.*

**Response: Thanks for the comment. Both of GlobColour and OCCCI (from Plymouth Marine Laboratory) incorporate SeaWiFS, MERIS, MODIS-AQUA, and VIIRS. The same components make the two products kind of similar. However, the uncertainty information and algorithms are not the same (http://www.globcolour.info/CDR_Docs/GlobCOLOUR_PUG.pdf). The information is added in the manuscript (P9, L2-3).**

*11. P9, L21. Suggest to remove this paragraph as it was already mentioned in P8, L20.*

**Response: Thanks for your comments. This paragraph is removed from section 3.1.**

*12. P10, L4. In addition to these physical processes, iron limitation is another main reason for high macronutrient levels in this region, which should be mentioned here.*

**Response: Thank you for your comments. The effect of iron limitation is emphasized in the revised manuscript (P11, L22-23).**

*13. P10, L10. Refer to the figure (Fig. 1 c,f,i) at the end of the first sentence.*

**Response: Thank you for your comment.  The sentence "some noticeable discrepancies…." is referred to Fig. a3, b3, c3 (P12, L2).**

*14. P10, L22 to P11, L4. This paragraph belongs to the methods section.*

**Response: Thank you for the comment. The paragraph is moved to the analysis method section.**

*15. P11, L2 to P12 L3. Again this paragraph belongs to the method section. Also, mention what the half-saturation constant is to set to in the model for each nutrient.*

**Response: Thank you for the comment. The paragraph is moved to the methodology section. The coefficient K is not a constant, but parameterized based on minimum half-saturation constant and concentrations of nutrients, phytoplankton, and diatoms. This is mentioned in the revised manuscript and references are added (P10, L11-13).**

*16. P12, L21. I don't think the chlorophyll levels are high for the equatorial Pacific and the Southern Ocean. They may be "relatively" high compared to the surrounding seas, but the absolute magnitudes are low.*

**Response: Thanks for the comment. The description is revised accordingly (P13, L24-27).**

*17. P13, L7. I am not sure what it means by "the intermediate concentration regions". Maybe provide a number?*

**Response: Thanks for the comment. Here the intermediate concentration refers to ~0.5 mg Chl m$^3$. The manuscript is revised accordingly (P13, L24).**

18. *P13, L11. Instead of the International Date Line, refer to the longitude coordinate? Not every reader knows the exact location of the date line. This and the next sentence can be easily identified if the model-obs subplots are provided.*

**Response: Thanks for the comment. The International Date Line refer to 180°E. Also, the model-obs subplots for figure 5 and 6 are provided (P15, L14-15).**

19. *P13, L14. From 1998? Fig. 6 says from 2003.*

**Response: Thank you for the comment. The NPP is from 2003 to 2014. The manuscript is revised accordingly (P15, L12).**

20. *P13, L25. PAR should be defined earlier in the text, and here just write as PAR.*

**Response: Thank you for the comment. PAR is defined in the model description section (P5, L19).**

21. *Figure 6. Caption and the figure text do not match. Please check more carefully. In the caption, b = VGPM, c = Eppley-VGPM, d = CbPM, whereas in the figure, b = VGPMMODIS, c = CbPM-MODIS, d = Eppley-MODIS.*

**Response: Thank you for the comment. The figure text is right, and the manuscript is revised.**

22. *P14, L25. Here and elsewhere, the term "deviation" is used to refer to the model-obs difference. This is a bit confusing because standard deviations are also used in the later analysis (e.g. P16, L2). Perhaps, use "difference" instead of "deviation".*

**Response: Thanks for the comment. We use difference and bias instead of deviation to refer to model-obs difference in the revised manuscript.**

23. *P15, L10. Again, this is where having the model-obs difference subplot would be helpful to support this sentence.*

**Response:Thanks for the comment. The model-obs difference subplot of alkalinity and DIC are added.**

24. *P16, L9. "due to the 3-dimensional correction : : :", unclear what this means. Add one or two sentences to explain.*

**Response: Thank you for the comment. The 3-dimensional correction refers to that the global inventory of nutrient and alkalinity are restored toward the observations once a year. We add few explanation and related reference (P18, L6-8).**

*25. P16, L12. Provide a reference for the observation value quoted here.*

**Response: Thank you for the comment. The reference is provided (P18, L10).**

*26. Table 1. As noted in the caption, the pre-industrial years between NESM v3 and IPCC AR5 differ by 100 years. Does this explain why the IPCC AR5 value is higher than NESM v3 because the former incorporates additional 100 years for the cumulative quantity? Also, in the caption, describe what the plus/minus values represent. P17, L25. Should this paragraph have its own section? It is beyond oceanic co2 uptake (Sec 3.4).*

**Response: Thank you for the comment.  The former additional 100 years can only explain very small part of the higher value, because the atmospheric $CO_2$ concentration change during this period is small. In another simulation started from 1750 (not shown in this manuscript), the total cumulative quantity is 141.7 Pg C.**

**The plus/minus values are the uncertainty range provided by IPCC AR5 (P39).**

**We put this paragraph in a new section named "Assessment by Taylor diagram".**

*27. P18, L21. ", which is associated : : : deep ocean," this middle block of sentence is not supported by any figure or reference, and also is unnecessary. It can just be removed and simply be stated as "The reduction of mixed layer depth indicates a more stratified upper ocean : : :".*

**Response: Thank you for the comment. The manuscript is revised accordingly (P20, L18).**

*28. Sec 3.5.1. This section only discusses the results of the FC simulation. Maybe provide some comments on the different simulations (e.g. BC vs RC simulations).*

**Response: Thank you for the comment. SAT, MLD, and AMOC changes in RC and FC simulations are almost the same, while those changes in the BC simulation are relative small. We add some comments on the BC and RC simulations in the revised manuscript (Section 3.6.1).**

*29. P19, L11. This first paragraph is already mentioned in the previous section, so why repeat here?*

**Response: Thank you for the comment. This paragraph is removed from Section 3.6.2 in the revised manuscript.**

*30. P21, L21 to P22, L8. This block of paragraphs belongs to the Method section. Having this much of methodological details in the results section breaks the flow of the paper. Please consider moving it to the Method section.*

**Response: Thanks for the comment. We move this paragraph to the Method section.**

31. *P22, L16. "Therefore, : : :" this sentence is unclear to me. Especially for the carbon climate parameter, which can be both positive and negative as shown in Figure 16d. Do the authors mean that it is negative in the year 2100?*

**Response: Thanks for the comment. During the former several decades, the interannual variation of oceanic $CO_2$ uptake is of the same magnitude of carbon-climate sensitivity resulted oceanic $CO_2$ uptake. This is why we can see positive carbon-climate parameter in the first several decades in Figure 16d. Here we refer to the parameters in the year 2100. We add "in the year 2100" in the revised manuscript (P23, L11).**

32. *P23, L3. This paragraph could move into the Method section and be combined with the block of paragraphs describing the sensitivity parameter derivation. Also, "4xCO2" is unclear.*

**Response: Thanks for the comment. We move this paragraph to the method section. 4XCO2 means the atmospheric $CO_2$ concentration quadrupled, and the description is added.**

33. *P24, L12. "Our results suggest : : :" How does your result support that temperature-dependence is necessary?*

**Response: Thanks for your comments. We delete this sentence in Conclusion Section, but add a brief explanation in Section 3.2 (P14, L22-25).**

**Temperature-dependence is necessary to produce the meridional distribution of NPP. When temperature-dependence is considered (NESM v3, CBPM-MODIS, and EPPLEY), we get high level NPP in low-latitude ocean and low level NPP in high-latitude oceans, while it is opposite when temperature-dependence is unconsidered (VGPM).**

34. *P25, L1. "The strong : : :", but why does NESM v3 overestimate nutrients? The iron limitation is too strong? Too strong vertical mixing? A few speculations can be helpful.*

**Response: Thanks for the comment. Figure 4 indicates the strong iron limitation and the bias of preformed phosphate in Figure 3 indicates a strong vertical mixing. The possible causes for the overestimated nutrients is provided in the revised manuscript (P25, L25-P26, L2).**

35. *P25, L10. Why do the results of IPSL-CM5A-LR appear here and not in the results section? Also, why choose this particular CMIP ensemble member over others? I don't really see the point of adding the comparison with IPSL-CM5A-LR.*

**Response: Thanks for the comment. We remove the results of IPSL-CM5A-LR from Conclusion Section, but add some comparisons to Section 3.4 (Taylor diagram). Also, a detailed comparison with IPSL-CM5A-LR is provided in the supplementary material.**

**The atmospheric component and oceanic component of IPSL-CM5A-LR and NESM v3 are both adopted from ECHAM and NEMO. This makes the two model kind of similar, and some readers wonder how different between the simulation results from the two models.**

*Editorial corrections*

*P1, L15. "total CO2 uptake" –> Use subscript for "2"*

*P3, L2. "; 2)" –> "; and 2)"*

*P3, L4. "the effect of CO2 concentration" –> "the effect of increasing atmospheric $CO_2$ concentration"*

*P4, L7. Should "NUIST-CSM-2.0.1" be "NESM v3"?*

*P4, L16 (and elsewhere). Add a space between 10 and m (10 m instead of 10m).*

*P5, L18-19. ": nanophytoplankton and diatoms," –> "(nanophytoplankton and diatoms)" and similarly for zooplankton.*

*P5, L22. "photosynthetic" –> "photosynthetically"*

*P6, L9. POM is already defined in P5, L25.*

*P6, L9. "diatoms silicate", should this be "biogenic silica"?*

*P6, L9. "described by : : : corresponding to" –> "is partitioned into"*

*P12, L2. "when all nutrients : : : than 0.9." –> "when the annual mean nutrient coefficients are greater than 0.9 for all nutrients."*

*P12, L14. SeaWiFS should be defined in P9, L1. OCCCI was already defined in P9, L2.*

*P12, L19. "plankton" –> "phytoplankton"*

*P14, L23 Add a space between alkalinity and are.*

*Fig. 7. Superscript for alkalinity units. Also add a space between from and 1985.*

*P14, L23. Add a space between alkalinity and are.*

*P15, L9. Add a space between ocean and means.*

*P16, L2. I don't think SD has been defined previously. If so, define here.*

*Figure 10. In the figure, the last subplot is labelled as "c) GLODAP v1", which should be "d) GLODAP v1".*

*P16, L24. "receptively" –> "respectively".*

*Figure 12. Caption: "statistical patterns" –> "spatial patterns"(?); "carbon-related" –> "biogeochemical"; "upper ocean" –> "upper 100-m ocean". Provide the information on observations for nutrients, DIC, and alkalinity, such as done for NPP and chlorophyll.*

*P18, L3. "to atmospheric" –>"to increasing atmospheric".*

*P18, L5-L8. "presented" –> "present" and "compared" –> "compare".*

*P18, L20. "and acting to mitigate" –> "and".*

*P18, L21. "MLD is seen decreasing" –> "Modeled MLD decreases".*

*P18, L24. "is seen in" –> "is projected for".*

*Figure 15. In the caption "FC-RC-BC" –> "FC-RC+BC"?*

*Figure 16. Caption "cumulated" –> "cumulative". "atmospheric co2 (c) and : : :" –> "(c) atmospheric co2 and (d) : : :"*

*P21, L25. ". The" –> ", the". Similarly for P22, L4.*

*P23, L11. "estimated by CMIP5 models range" –> "estimated for CMIP5 models ranging"*

*P24, L3. "Mid-Eastern" –> "Eastern"*

*P24, L20. "nonlinear of " –> "nonlinear response of oceanic"*

*P25, L16. "precipitation in" –> "precipitation simulated in"*

*P25, L18. "which would lead" –> "which leads"*

*P25, L21. "NUIST-CSM", should this be "NESM v3"?*

**Response: Thank you so much for pointing out these detailed editorial corrections. All above editorial corrections are checked and revised accordingly.**

*P24, L14. The numbers quoted in this paragraph is inconsistent with the ones appeared in the Results section. Specifically, year 2016 (should this be 2011? Table 1), 149, 150 +/- 20, and 0.8. Please check these numbers with the Results section.*

**Response: The result from the pre-industrial era (1850) to the year 2011 are compared with IPCC AR5, while the cumulative uptake from 1870 to 2016 are compared with recent results from Le Quéré et al. (2018). We give both comparisons in the manuscript (P18, L14-22).**

**Comments from reviewer2**

**General comments**

*Dai and co-authors evaluate the ability of their earth system model NESM v3 to represent the carbon cycle (and, particularly, the CO2 uptake) and the representation of several marine biogeochemical tracers (nutrients, alkalinity, DIC, chlorophyll and net primary production). NESM v3 performances are compared with observations and, occasionally, to CMIP5 models. As regional discrepancies are identified, the authors discussed their physical (e.g. weak upwelling in the Indian Ocean, strong convective mixing at high latitudes and more generally, shortcomings in simulated ocean circulation) or biogeochemical (iron limitation in the Southern Ocean, excessive remineralization in the deep Northern Pacific) origins.*

*2) Relevance of the subject*

*Such a paper evaluating the limits of a modelling platform can be very useful to the scientific community which is going to use and analyse NESM v3 outputs, especially if the model has contributed to the CMIP6 Intercomparison Project. However I did not really understand from the text if the model described and used in this paper has really been a part of CMIP6: p.3, l.10: "as a registered model of CMIP6" but p.7, l.21: "following the protocol of CMIP5" Maybe the authors may explain why not using the protocol of CMIP6 (to be CMIP6 fully compliant) ?*

**Response: Thank you for you positive comments on this manuscript. In this study, historical and SSP5-8.5 experiments follow the CMIP6 protocol, of which the parameters are differ from CMIP5 protocol, including solar forcing, aerosol, land use, and so on. Then we compared these results with observations and CMIP5 model results to evaluate the paper.**

**However, the only differences of 1ptCO2 experiment between the protocol of CMIP5 and CMIP6, is that CMIP5 only requires 140 years integration, while CMIP6 requires more than 150 years. Considering that we compared the carbon feedback parameter diagnosed from the 1ptCO$_2$ experiment with CMIP5 model results, we only show 1ptCO2 for 140 years in this manuscript, e.g. following CMIP5 protocol.**

*3) General structure*

*The readability could be improved by a better structure. Please have a look at Séférian et al. (2019, https://doi.org/10.1029/2019MS001791) which provides an evaluation of CNRM earth system model for CMIP6 by comparing it to observations, as well as to CMIP5 multi-model ensemble, and to an earlier version of the same model. Keeping the introduction in its actual state (i.e. focusing on carbon uptake), I suggest to move the description of NESM v3 found in the introduction (p.3, l11-16) to the dedicated section (2.1.1). But I would rather expect a more focused introduction, relaying previous/other model evaluations of the carbon cycle and uptake. As it is, the scope of the introduction is a bit too wide.*

**Response: Thank you for the comment. We move the description of NESM v3 to section 2 and clarify our objective in this section (P3, L11-14).**

*4) Results*

*I suggest to discuss the magnitude of the nutrient biases obtained in section 3.1 in regard of those obtained with other models like CMIP5 models (maybe a short summary of the published CMIP5 literature on these aspect will*

*be enough). This would help the reader to know how NESM v3 places itself in the CMIP models diversity. This is also true for the section relative to the Taylor diagram (Fig. 12): please see my specific comment.*

**Response: Thank you for the comment. The comparison with CMIP5 models are added to section 3.1 and Taylor diagram.**

*The section 3.5.2 discussing the coupling between the "radiative" (i.e. in this case only atmospheric radiation is affected by changing concentrations of atmospheric CO2) and "biogeochemical" (i.e. in this case only the ocean carbon cycle is affected by changing atmospheric CO2) sensitivity experiments is quite interesting. However if the motivation of this paper is to evaluate model skills in modeling carbon-related biogeochemical species, the study of the non-linearity of their sum appears a bit beyond the lines of the paper. I would recommend that either the authors restructure (a bit) the current draft or clarify the aim of their study.*

**Response: Thanks for the comment. We clarify the aim of the study in Introduction Section (P3, L11-14).**

**For oceanic $CO_2$ uptake, we evaluate both the amount and the sensitivity. In this study, the sensitivity is represented by the two parameters and their nonlinearity. The section 3.6.2 is a bit restructured and we want to show how the sensitivity and nonlinearity of oceanic CO2 uptake distribute and originated in the NESM v3 (P21, L6- P22, L26).**

*5) Discussion*

*I would expect of a paper aiming at evaluating a model that the "Discussion and Conclusion section" would give more details of how this model behaves (in terms of modeled carbon cycle and CO2 uptake here) in comparison with other models or in the context of the other CMIP models. If such comparisons are occasionally done in the current draft version, it would be valuable to systematise them.*

**Response: Thanks for the comments. The Discussion Section is restructured, and the comparison between the NESM v3 and other models is summarized in an individual paragraph (P25, L9-18).**

*6) Language*

*I would recommend a careful reading which may easily help to correct the typing errors.*

**Response: Thanks for the suggestion. We checked throughout the manuscript.**

*7) Specific comments*

*p.3, l.19: modes -> models*

*p.4, l.4: includes -> that includes or including*

*p.4, l.15: tripole -> tripolar grid*

*p.9, l.18: the modeled result -> the modeled sea-air CO2 fluxes to a 4◦x5◦ grid.*

*p.10, l.3: Fig. 11 -> Fig. 1 ?*

*p.14, l.23: alkalinityare -> alklinity are*

*p.16, l.11: ",", -> "."*

*p.16, l.24: receptively -> respectively*

*p.17, l.4: stimulated -> simulated ?*

*p.18, l.20: MLD is seen decreasing -> "MLD is seen to decrease" sounds to me more correct.*

*p.21, l.3: "some regions of the Northern Atlantic even appear CO2 outgassing". The formulation sounds weird to me, please check its grammatical correctness.*

*p.25, l.27: ". . ." -> "."*

**Response: Thanks for the comments. All the mentioned editorial errors are revised.**

*p.7-8, l.27, l.1-4: "To separate the effect of atmospheric CO2 and global warming on the ocean carbon cycle, we performed three types of experiments (biogeochemically coupled, radiatively coupled, and fully coupled). These types of simulations were also performed by previous studies that investigated the effect of CO2 and global warming on the global carbon cycle (Friedlingstein et al., 2006; Arora et al., 2013; Schwinger et al., 2014)."*

*I suggest to slightly reorganize the above paragraph in order to properly introduce the list items that follows. I suggest something like that:*

  *"Following Friedlingstein et al., 2006; Arora et al., 2013; Schwinger et al., 2014, we performed three types of experiments (biogeochemically coupled, radiatively coupled, and fully coupled) to separate the effect of atmospheric CO2 and global warming on the ocean carbon cycle: 1) Biogeochemically coupled (BC). . .."*

**Response: Thank for the comment. The above paragraph is rewritten accordingly in Section 2.2 (P7, L23).**

*p.11, l.21-22: why is the ocean circulation so different in IPSL-CM5A-LR and NESM, as both models share the same oceanic model (NEMO) ?*

**Response: Thanks for the comment. We compared more than 20 different ocean physical climate fields, including temperature, salinity, wind stress, ocean current, and MOC, between NESM v3 and CMIP5 models. The results show some differences between NESM v3 and IPSL-CM5A-LR. I think the differences possibly come from several sources, including different coupler strategy, atmospheric and sea-ice component, and unique modifications in the ocean model of NESM v3. Some of the modifications are mentioned in the Cao et al. (2018).**

**However, we have not exanimated how these differences affect biogeochemical cycle. We remove this sentence from discussion section.**

*p.17, l.23: Similar to the vertically integrated inventory (Fig. 10): I suggest to add the figure you are referring to, in order to facilitate the reading. I also suggest to clarify (in the text or, at least, in the caption of Fig. 11) the*

*period on which DIC has been averaged for computing these vertical sections: I guess that these vertical sections have been averaged between 1985 and 2014 as other vertical sections ? As it gives an indication of the temporal persistence of the bias, I would suggest to mention it. The vertically integrated inventory of Fig. 10 are not representative of the whole period but represent only few years around 2002 or 1994.*

**Response: Thanks for the comment. The figure referred is added in the manuscript (P19, L22). For figure 12, the vertical sections of anthropogenic DIC are averaged from 2000 to 2004 to get a better comparison with observations, because the observation of DIC in GLODAP v2 is normalized to the year of 2002.**

*p.17, l.25 : it would be very helpful to add other models data or even CMIP5 ensemble mean on this Taylor diagram for all the biogeochemical fields in order to get an idea of how NESM v3 behaves in the current modeling landscape.*

**Response: Thanks for the comment. We add simulation results from IPSL-CM5A-LR to the Taylor diagram, and add descriptions of other CMIP5 models' performance by reviewing previous evaluation papers (Section 3.4, P17, L2-18).**

*p.26, l.15-16: please replace XXX and YYYY by providing publication numbers, or delete the sentence.*

**Response: Thanks for the comment. This part will be revised in final revision.**

*8) Figures*

*p.40, Figure 2: to improve the readability of this figure, I suggest to separate the 3 oceanic regions (Atlantic, Pacific, Global) and to increase the tick labels and titles. You can keep all the 3x6 subplots on the same figure, but at least try to increase the margin below the 6 subplots related to the Atlantic ocean, and also increase the margin below the 6 subplots related to the Pacific ocean following a pattern like:*

*a0 b0. c0*

*a1. b1. c1*

*<- increased margin*

*a2. b2 c2*

*a3. b3. c3*

*<- increased margin*

*a4. b4. c4*

*a5 b5. c5*

*p.41, Figure 3: same recommendation than for Figure 2, please add an increased margin between the cluster of subplots for preformed PO4 and the cluster of subplots showing regenerated PO4. Increase all ticks labels and titles. As the whole analysis of Fig. 3 is based on biases values, it would be very helpful to show these biases on Fig. 3. You could show a first line of vertical sections relative to observations and then a second line with biases (model-obs) rather than model means.*

**Response: Thanks for the comment. Figure 2 and 3 are revised accordingly.**

*p.42, Figure 4: please mention the source of these nutrients limitation patterns. Are they diagnosed from NESM v3 model ?*

**Response: Thanks for the comments. The limitation patterns are diagnosed from fully-coupled simulation of NESM v3. The figure caption is revised accordingly.**

*P.44, Figure 6: be careful with the mismatch between subplots titles and their name in the legend.*

**Response: Thanks for the comment. The caption of figure 6 are revised accordingly.**

*p.46, Figure 8: see recommendations for Figures 2 and 3.*

**Response: Thanks for the comment. The ticks labels and titles are increased, as well as the margin below the six subplots. We also replace model distribution with the bias map.**

*p. 51 and 53, Figures 13 and 15: please add units of analysed fields.*

**Response: Thanks for the comment. The units are added in the figure caption.**

*9) Bibliography*

*Séférian, R., Nabat, P., Michou, M., Saint-Martin, D., Voldoire, A., Colin, J., et al (2019). Evaluation of CNRM Earth-System model, CNRM-ESM2-1: role of Earth system processes in present-day and future climate. Journal of Advances in Modeling Earth Systems, 11. https://doi.org/10.1029/2019MS001791*

[revised manuscript text omitted]

a1) Phosphate_Model a2) Phosphate_Observation a3) Phosphate (Model-Observation)
b1) Nitrate_Model b2) Nitrate_Observation b3) Nitrate (Model-Observation)
c1) Silicate_Model c2) Silicate_Observation c3) Silicate (Model-Observation)

[Figure]

**Figure 1.** Annual mean (averaged over 1985 to 2004) upper ocean (averaged in the upper 100 m) distribution of phosphate (a1, a2), nitrate (b1, b2) and silicate (c1, c2) from the NESM v3 simulations (FC) and the WOA18 observation dataset (in the unit of mmol m⁻³). The difference between model simulation and observation are also shown (a3, b3, and c3)

[Figure]

[Figure]

**Figure 2.** The latitude-depth distribution of silicate (a), phosphate (b), and nitrate (c) averaged from 1985 to 2014 from the NESM v3 simulation (FC) compared withand the WOA18 observation dataset (with a unit of mmol m$^{-3}$). a, b, and c represent the silicate, phosphate, and nitrate, respectively. 0a1, a2, b1, b2, c1, and 1c2 represent the distributions in the Pacific Ocean, 2a3, a4, b3, b4, c3, and 3c4 represent the distributions in the Atlantic Ocean, and 4a5, a6, b5, b6, c5, and 5c6 represent the distributions in the Global Ocean.

[revised manuscript text omitted]

---

## Author Response (AR2)

**Response to editor's comments**

We would like to thank you for the in-depth review and constructive comments. Below we provide the point-to-point responses to each comment. Specific comments are given in italic and responses are given in bold. Manuscript with revisions marked using track changes is also provided.

**Specific Comments**

*1) Please can you revise your discussions section, to ensure that ALL of the statements you make are backed up with evidence (i.e. at present there are lots of statements which are probably underpinning by your results section, but there is no reference back to the figure etc. which presents that evidence).*

10 **Response: Thanks for the comment. The corresponded references and figures are provided.**

*2) Please can you also significantly expand the paragraph your have added P25 L9-18. The reviewer sensibly asks for this comparison, but your response is much too light.*

**Response: Thanks for the comment. This paragraph is a bit reconstructed to include the comparison of**
15 **biogeochemical fields, the amount of oceanic carbon uptake, and the sensitivity of oceanic carbon uptake. A summary of the comparison of biogeochemical fields between the NESM v3 and IPSL is given and we also add the comparison of oceanic carbon uptake in the historical period between the NESM v3 and CMIP5 models.**

20 *3) Finally, please undertake a (hopefully) final proof-read and edit to ensure that throughout the manuscript the subject of each sentence is completely clear. For example, you have revised the final paragraph of the manuscript to:*
*'Overall, compared with both observations and CMIP5 models, the NESM v3 does a good job in simulating ocean biogeochemical fields and oceanic carbon uptake. Despite these model-observation*
25 *discrepancies, it is expected that NESM v3 can be used as a useful modeling tool to study interactive feedbacks between the ocean carbon cycle and climate change and the underlying mechanisms.'*
*The second sentence says 'Despite these model-observation discrepancies...' but the preceding sentence (and consequently nowhere in the paragraph it) makes no reference to model-observation discrepancies.*
**Response: Thanks for the comments. We check throughout the manuscript sentence by sentence to make the**

**writing clear. The publication number in Acknowledgement which is now shown as XXXX, will be given when the manuscript is in final production.**

[revised manuscript text omitted]